# Systemic acquired resistance networks amplify airborne defense cues

Marion Wenig [1], Andrea Ghirardo [2], Jennifer H. Sales[1], Elisabeth S. Pabst[1], Heiko H. Breitenbach[1], Felix Antritter[2], Baris Weber [2], Birgit Lange[1], Miriam Lenk[1], Robin K. Cameron [3], Joerg-Peter Schnitzler [2] & A. Corina Vlot [1]

Salicylic acid (SA)-mediated innate immune responses are activated in plants perceiving volatile monoterpenes. Here, we show that monoterpene-associated responses are propagated in feed-forward loops involving the systemic acquired resistance (SAR) signaling components pipecolic acid, glycerol-3-phosphate, and LEGUME LECTIN-LIKE PROTEIN1 (LLP1). In this cascade, LLP1 forms a key regulatory unit in both within-plant and between-plant propagation of immunity. The data integrate molecular components of SAR into systemic signaling networks that are separate from conventional, SA-associated innate immune mechanisms. These networks are central to plant-to-plant propagation of immunity, potentially raising SAR to the population level. In this process, monoterpenes act as microbe-inducible plant volatiles, which as part of plant-derived volatile blends have the potential to promote the generation of a wave of innate immune signaling within canopies or plant stands. Hence, plant-to-plant propagation of SAR holds significant potential to fortify future durable crop protection strategies following a single volatile trigger.

[1] Helmholtz Zentrum Muenchen, Department of Environmental Science, Institute of Biochemical Plant Pathology, Ingolstaedter Landstr. 1, 85764 Neuherberg, Germany. [2] Helmholtz Zentrum Muenchen, Department of Environmental Science, Institute of Biochemical Plant Pathology, Research Unit Environmental Simulation, Ingolstaedter Landstr. 1, 85764 Neuherberg, Germany. [3] McMaster University, Faculty of Science, 1280 Main St. West, Hamilton, ON L8S 4K1, Canada. Correspondence and requests for materials should be addressed to A.C.V. (email: corina.vlot@helmholtz-muenchen.de)

Plants ward off pathogens via immune mechanisms that rely on a distinct set of phytohormone signaling pathways. One of the main immunity-related phytohormones is salicylic acid (SA), which is essential for defense in plants against (hemi-) biotrophic pathogens[1]. It acts downstream of the recognition of pathogen elicitors or pathogen-associated molecular patterns (PAMPs) leading to PAMP-triggered immunity (PTI) and of pathogen effectors leading to effector-triggered immunity (ETI)[2]. Both PTI and ETI trigger long-distance signaling resulting in the induction of systemic acquired resistance (SAR), an SA-dependent induced resistance in distal uninfected plant parts that acts against a broad range of pathogens[1,3]. One of the first long-distance signals that was proposed to propagate SA-dependent immunity in distal tissues was the volatile derivative of SA, methyl salicylate (MeSA)[4]. In SAR, MeSA is believed to act as a phloem-mobile signal that is hydrolyzed in the systemic tissue to its bio-active derivative SA to trigger resistance[4]. In the same way, MeSA might act as an airborne cue inducing pathogen resistance in different plant species[5–7]. Notably, MeSA also positively influences indirect defenses of plants against herbivores by attracting herbivore natural enemies, for example in tomato[5,8].

In addition to MeSA, both the non-protein amino acid pipecolic acid (Pip) and its SAR bio-active derivative N-hydroxypipecolic acid are essential for SAR and appear to be systemically mobile[9–12]. Recent evidence places Pip upstream of a cascade of SAR-associated compounds, including nitric oxide (NO) and reactive oxygen species ($H_2O_2$)[12]. The $C_9$ dicarboxylic acid azelaic acid (AzA) accumulates downstream of NO and $H_2O_2$ and acts upstream of glycerol-3-phosphate (G3P) in the same pathway[13–15]. G3P, in turn, promotes SAR in a positive feedback loop with the predicted lipid transfer proteins AZELAIC ACID INDUCED1 (AZI1) and DEFECTIVE IN INDUCED RESISTANCE1 (DIR1)[15,16]. Besides Pip, N-hydroxypipecolic acid, and MeSA, putative long-distance SAR signals include AzA, G3P, DIR1, DIR1-LIKE, the diterpenoid dihydroabietinal, and the volatile monoterpenes α- and β-pinene[4,12,14,17–20].

The importance of plant volatiles in immunity against insects, including plant-to-plant signaling, has been extensively studied[21–23]. Recently, we showed that SA-mediated immunity or SAR is similarly propagated between individual A. thaliana plants[20]. Emissions of the monoterpenes α-pinene, β-pinene, and camphene were induced during ETI and essential for SAR. Headspace exposure of plants to a mixture of α- and β-pinene induced gene expression changes related to SA signaling and SAR, suggesting that these monoterpenes act as signaling intermediates. Importantly, plant-to-plant experiments suggested the establishment of SAR in the receiver plants upon perception of monoterpene emissions from the sender plants[20]. Arabidopsis thaliana monoterpene emissions depend on the SA and SAR regulatory ENHANCED DISEASE SUSCEPTIBILITY1 (EDS1) gene[20,24]. The corresponding mutant also displays reduced accumulation of AzA and the putative carbohydrate-binding LEGUME LECTIN-LIKE PROTEIN1 (LLP1)[24,25]. LLP1 is essential for SAR and promotes systemic immunity most likely in parallel with SA signaling[24].

Collectively, the data suggest regulatory interactions between different SAR-associated signals. However, it is unclear how different SAR-associated signaling components converge to induce systemic immunity[26,27] and how systemic signals are transmitted. Similarly to SA[28,29], Pip appears to be systemically mobile[11,12]. However, the establishment of SAR requires SA and Pip accumulation in systemic and not in primary infected tissue[12,30], suggesting that their long-distance transport is not relevant for SAR. In contrast, petiole exudate experiments have shown that G3P, DIR1, and AZI1 are required for local SAR signal generation or transmission[13,16,18,31], suggesting a role of these SAR-associated signals in phloem-mediated transfer of immune signaling, while AZI1 concomitantly might play a role in SAR signal transduction in the systemic tissue[28]. Here, we establish signaling connections between Pip, G3P, LLP1, and volatile monoterpenes, connecting local to systemic tissues and identifying a key role of the SAR signaling network in the propagation of volatile defense cues. The data suggest an unexpected, airborne mode of SAR signal transfer and a possible ecological relevance of the plant internal SAR signaling network in fortifying the innate immune status of leaf canopies or plant populations.

## Results

**LLP1 promotes systemic responses to vascular SAR signals.** LLP1 is essential for SAR[24], and we aimed here to characterize the underlying mechanisms. First, we used petiole exudates to differentiate between local and systemic effects of LLP1 on SAR. In this and all following experiments SAR was induced by a local infection of A. thaliana with P. syringae pathovar tomato (Pst) carrying the effector AvrRpm1 and compared with a mock induction with 10 mM $MgCl_2$. Petiole exudates were collected from Pst/AvrRpm1-infected and mock-treated donor plants and subsequently infiltrated into leaves of naive recipient plants (Fig. 1a). One day (d) later, the infiltrated (recipient) leaves were either harvested or infected with Pst. Petiole exudates from infected wild-type (wt) donor plants induced eightfold more PR1 transcript accumulation in wt recipient plants as compared with petiole exudates from mock-treated wt donors (Fig. 1b). Similarly, petiole exudates from infected llp1-1 donors induced PR1 transcript accumulation in wt recipients, albeit to a lesser extent compared with petiole exudates from infected wt donors (Fig. 1b). Thus, LLP1 is not essential for, but might contribute to SAR signal generation. In contrast, petiole exudates from infected wt donors did not change PR1 transcript accumulation in llp1-1 recipients (Fig. 1b), suggesting that LLP1 is essential for SAR signal perception or propagation in systemic (recipient) leaves. To validate this hypothesis, Pst titers were monitored 4 days post-inoculation (dpi) in recipient plants. As expected, petiole exudates from infected wt donors enhanced resistance of wt recipients to Pst growth compared with petiole exudates from mock-treated wt donors (Fig. 1c). Petiole exudates from infected llp1-1 donors induced a similar reduction of Pst growth as those collected from infected wt donors (Fig. 1c). Reciprocally, petiole exudates from infected wt donors did not enhance resistance of llp1-1 recipients to Pst growth, indicating that LLP1 is essential for the recognition of or downstream responses to SAR signals in systemic (recipient) tissues (Fig. 1c).

*LLP1* has several close homologs in the A. thaliana genome, including *LLP2* (At3g16530), and *LLP3* (previously referred to as *LECTIN*[32])[24]. The *LLP2* and *LLP3* coding sequences respectively share 77 and 76% sequence homology with *LLP1*, and it was therefore conceivable that the encoded proteins could be functionally redundant with LLP1. To investigate this, we generated RNAi lines in the Col-0 wt background that supported reduced transcript accumulation of *LLP1, LLP2*, and *LLP3* (*RNAi:LLP1-3*; Supplementary Fig. 1A). Similar to llp1-1 mutants, *RNAi:LLP1-3* did not respond with reduced Pst growth to petiole exudates from infected wt donors (Fig. 1d and Supplementary Fig. 1B). In contrast to petiole exudates from infected llp1-1 donors, petiole exudates from infected *RNAi:LLP1-3* donors did not enhance resistance of wt recipients to Pst growth. Whereas we cannot exclude co-silencing of additional similar genes, the data suggest an additional or redundant role of one or more LLP in SAR signal generation.

**LLP1 acts downstream of pipecolic acid and volatile pinenes.** Because LLP1 appeared to act mainly in systemic SAR signal recognition or downstream responses, we tested which SAR signals

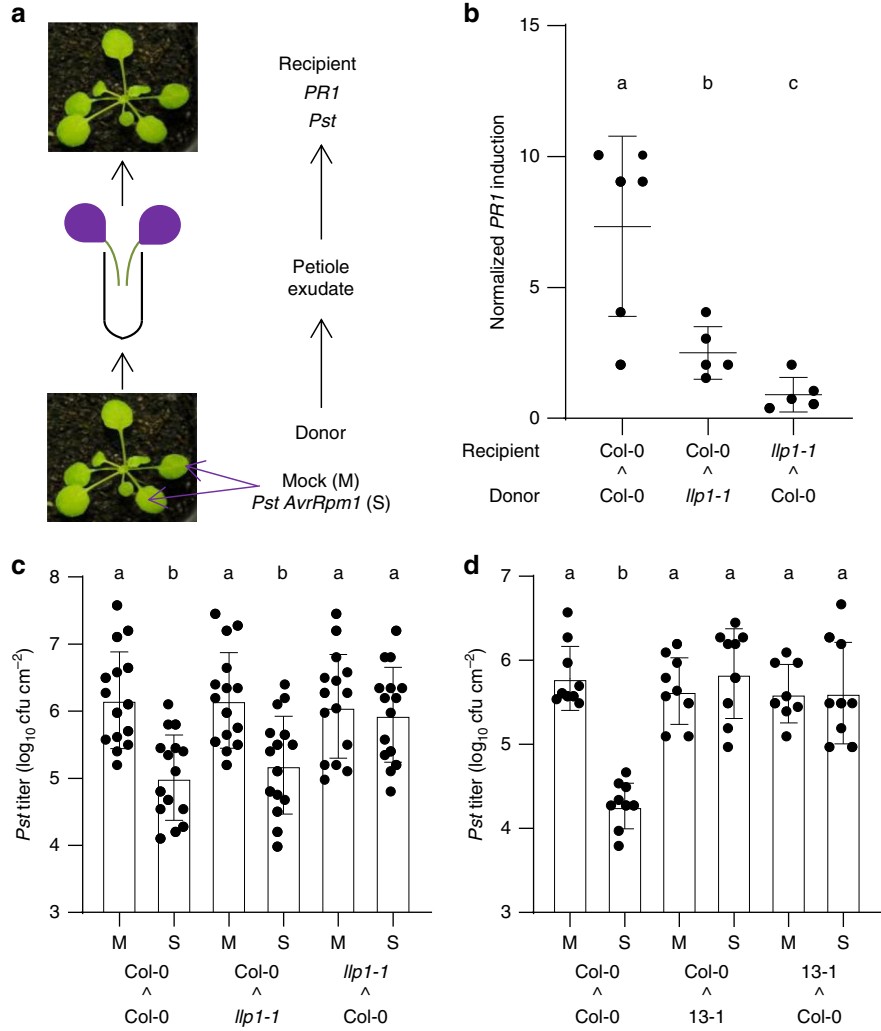

**Fig. 1** Legume lectin-like protein 1 (LLP1) is necessary for the recognition of or downstream responses to vascular systemic acquired resistance (SAR) signals. **a** Setup of a petiole exudate experiment. Leaves of donor plants were inoculated with *Pseudomonas syringae* pathovar *tomato* (*Pst*) carrying the effector locus *AvrRpm1* (SAR-induced; S) or mock-treated (M). Twenty-four hours later, their petiole exudates were collected and infiltrated into the leaves of naive recipient plants. **b** *Pathogenesis-Related 1* (*PR1*) transcript accumulation in recipients of petiole exudates from SAR-induced donor plants normalized to that in recipients of petiole exudates from mock-treated donor plants. Donor and recipient genotypes are indicated below the panel. Dots represent data from five to six biologically independent experiments; lines indicate average ± standard deviation. Grubb's outlier test identified statistically significant outliers in the data sets *llp1-1*-to-Col-0 and Col-0-to-*llp1-1*; these outliers were excluded from further analyses to assure normal distribution of the remaining data and are highlighted in gray in the source data file associated with this paper. **c**, **d** *In planta Pst* titers at 4 days post-inoculation (dpi) of the leaves of the recipient plants. The treatments of the donor plants are indicated below the bars. The donor and recipient genotypes are indicated below the panels; 13-1 refers to *RNAi:LLP1-3* line 13-1 (Supplementary Fig. 1A). Dots indicate individual results from three (**d**) to five (**c**) biologically independent experiments, including three replicates each. Bars represent the average of the indicated results ± standard deviation. Results with a second independent *RNAi:LLP1-3* line are presented in Supplementary Fig. 1B. **b**–**d** Different letters above bars indicate significant differences, one-way ANOVA, *P* < 0.05

might act upstream of LLP1. First, plants were exposed to volatile MeSA. After 3 days, the treated plants were inoculated with *Pst* and resulting *in planta Pst* titers were monitored at 4 dpi. Wt plants responded to MeSA exposure with reduced *Pst* growth as compared with the mock-treated control, and this response was independent of LLP1, 2, and 3 (Supplementary Fig. 2A). Infiltration of AzA into two lower leaves of *A. thaliana* enhanced resistance of systemic leaves to *Pst* and this response also appeared to occur independently of LLP1 (Supplementary Fig. 2B). In contrast, AzA-induced resistance was abolished in *RNAi:LLP1-3* plants, suggesting that LLP2 and/or LLP3, possibly together or redundantly with LLP1, contribute to downstream responses to AzA.

Next, we assessed the ability of *llp1* mutants to respond to Pip. Pip irrigation enhanced resistance of leaves of treated wt plants to *Pst* growth (Fig. 2a, b). Pip-induced immunity was abolished in

the *llp1-1* mutant, placing LLP1 downstream of Pip in SAR (Fig. 2b). In order to position LLP1 and also volatile monoterpenes along the Pip-associated SAR signaling cascade introduced above[12], we compared Pip-induced resistance with *Pst* in mutants that are associated with this pathway as well as in the monoterpene emission mutant *ggpps12* (formerly referred to as *ggr1-1*[20]). In contrast to the G3P-deficient *gly1-3* mutant, which did not respond to Pip irrigation with reduced *Pst* growth[12], *azi1-2* mutants responded to Pip with a wt-like reduction in *Pst* growth compared with that in mock-treated plants (Fig. 2c). Thus, the data uncouple AZI1 from G3P in the plant response to Pip. Similar to *llp1-1* plants (Fig. 2b), *ggpps12* plants did not mount a resistance response to *Pst* after Pip irrigation (Fig. 2c), suggesting that Pip responses depend on LLP1, G3P, and monoterpene biosynthesis or emission.

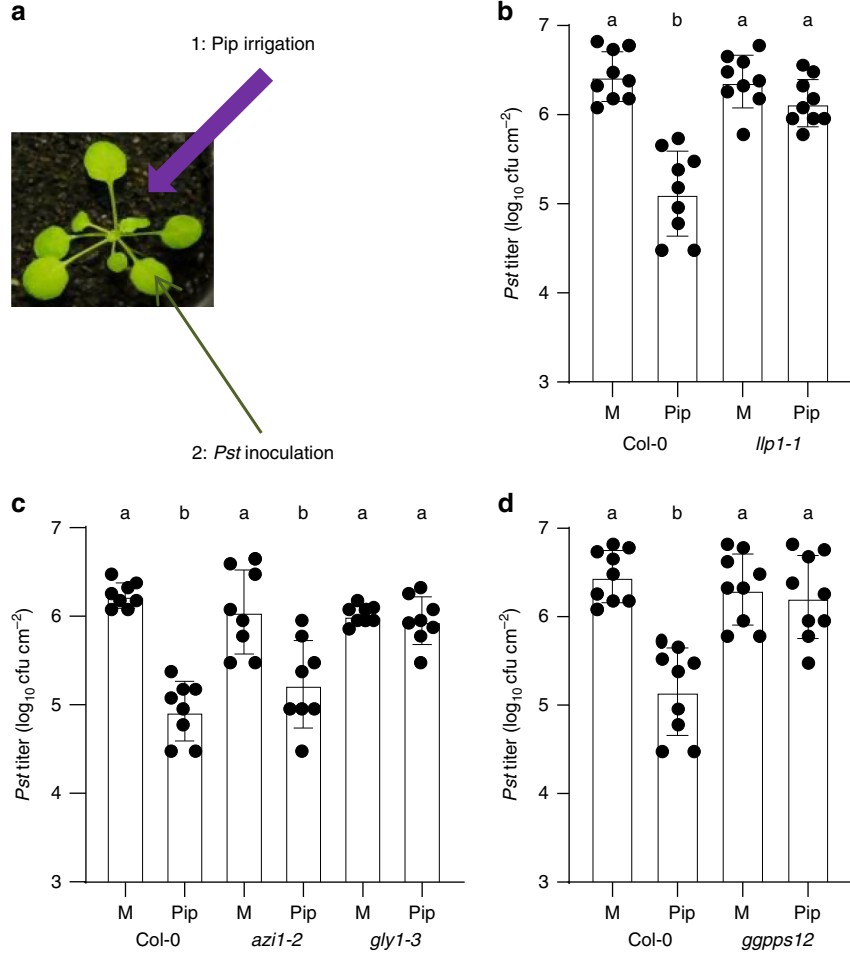

**Fig. 2** LLP1, glycerol-3-phosphate (G3P), and volatile monoterpenes act downstream of pipecolic acid (Pip) in Pip-induced resistance. **a** Setup of a Pip irrigation experiment. Plants were irrigated near the roots with Pip (or $H_2O$ as the mock (M) control) and subsequently inoculated in the leaves with *Pst*. **b**, **c** *In planta Pst* titers at 4 dpi of Pip- or mock-treated plants. The plant genotypes are indicated below the panels and include *gly1-3* with compromised G3P accumulation and *ggpps12* with compromised monoterpene biosynthesis. Dots indicate individual results from three biologically independent experiments per genotype and treatment (including three replicates each). Bars represent the average of the indicated results ± standard deviation. Different letters above bars indicate significant differences, one-way ANOVA, $P < 0.05$

In order to further define the positions of LLP1 and monoterpenes with respect to Pip in SAR-regulatory signaling networks, we monitored monoterpene-induced resistance in *llp1-1* and Pip-deficient *ald1* mutants (Fig. 3a). Exposure of wt plants to a mixture of volatile α- and β-pinene (molar ratio 1:1; Pin) for 3 consecutive days enhanced resistance of treated wt plants to *Pst* growth (Fig. 3b). Similar to Pip-induced resistance, Pin-induced resistance was abolished in the *llp1-1* mutant (Fig. 3b), placing LLP1 downstream of Pin in the establishment of systemic resistance. Resistance to *Pst* by exposure of plants to Pin was also abolished in the *ald1* mutant (Fig. 3b). The collective data suggest that Pip and Pin promote SAR either in two parallel signaling cascades converging on LLP1 or in one circular pathway together with LLP1.

**Pip accumulates upstream of LLP1, G3P, and monoterpenes**. In order to differentiate between parallel and circular signaling relationships between Pip and Pin, Pip accumulation and Pin emissions were monitored in *Pst/AvrRpm1*-infected compared with mock-treated plants. Pip accumulation was induced by *Pst/AvrRpm1* in wt plants and to a similar extent in *azi1-2*, *gly1-3*, *ggpps12*, and *llp1-1* mutants (Supplementary Fig. 3), supporting previous findings that Pip accumulates and/or acts upstream of

G3P and potentially AZI1 in the establishment of systemic resistance[12]. The data also place Pip accumulation upstream of monoterpene emissions and LLP1. Emissions of the mono-terpenes α-pinene, β-pinene, and camphene were induced by *Pst/AvrRpm1* in wt plants and to a lesser extent also in the *azi1-2* mutant (Fig. 4 and Supplementary Fig. 4). In contrast to basal emissions of these monoterpenes, the emissions of α-pinene, β-pinene, and camphene were not induced by *Pst/AvrRpm1* in *gly1-3*, *ald1*, and *llp1-1* plants (Fig. 4 and Supplementary Fig. 4). The collective data place Pip, G3P, and LLP1 upstream of mono-terpene emissions and hint at a possible circular mechanism involving these three SAR-associated signals acting upstream of AZI1.

**LLP1, G3P, and Pip promote monoterpene-mediated PTP immunity**. We recently showed that *A. thaliana* responds with enhanced *Pst* resistance to volatile emissions of *Pst/AvrRpm1*-infected (SAR-induced) neigbors (Fig. 5a, b)[20]. In these experi-ments, naive receiver plants were exposed to SAR-induced or mock-treated sender plants (Fig. 5a). After 3 days, the receivers were inoculated with *Pst* and the resulting *Pst* titers monitored at 4 dpi. Emissions from *ggpps12* mutants, which show reduced monoterpenoid biosynthesis, did not induce resistance in wt

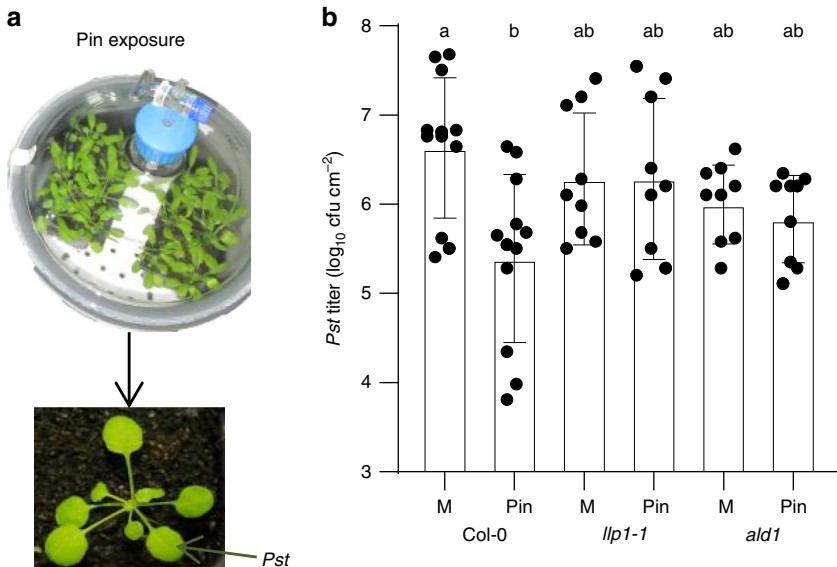

**Fig. 3** LLP1 and Pip act downstream of volatile monoterpenes in pinene-induced resistance. **a** Setup of a pinene fumigation experiment. Plants were exposed to a mixture of α- and β-pinene (molar ratio 1:1; Pin) in air-tight glass containers or to the solvent hexane as the mock (M) control treatment. The plants were subsequently released from the containers and inoculated in the leaves with *Pst*. **b** *In planta Pst* titers at 4 dpi of pinene (Pin)- or mock (M)-treated plants. The plant genotypes are indicated below the panel and include *ald1* with compromised Pip accumulation. Dots indicate individual results from three to four biologically independent experiments per genotype and treatment (including three replicates each). Bars represent the average of the indicated results ± standard deviation. Different letters above bars indicate significant differences, one-way ANOVA, $P < 0.05$

receivers, suggesting an important role of monoterpenes in plant-to-plant (PTP) transfer of innate immunity[20]. In contrast, a wt limitation of *Pst* titers was observed in *ggpps12* mutants exposed to emissions of SAR-induced compared with mock-treated wt plants (Fig. 5b). Thus, once perceived, monoterpenes do not appear to play a further role in establishment of downstream resistance responses.

Similarly to wt, SAR-induced *azi1-2* plants exhibited elevated α- and β-pinene emissions as compared with mock-treated plants (Fig. 4). When used as senders in PTP experiments, the resistance of wt receivers to *Pst* was enhanced in response to emissions from SAR-induced *azi1-2* plants (Fig. 5c). Furthermore, the immune response of *azi1-2* receivers to emissions of SAR-induced wt senders was compromised compared with that of wt receivers (Fig. 5c), suggesting an important role of AZI1 in the establishment of immunity in response to PTP cues.

Similar to *ggpps12* plants[20], *llp1-1*, *gly1-3*, and *ald1* plants emitted reduced levels of α-pinene, β-pinene, and camphene after infection as compared with wt plants (Fig. 4 and Supplementary Fig. 4). As a result, resistance in wt receivers to *Pst* was not enhanced in response to emissions of SAR-induced *llp1-1*, *gly1-3*, and *ald1* plants (Fig. 5d–f). Thus, the capacity of different SAR mutants to produce monoterpene emissions correlated with the presence of defense cues in their PTP volatile blends. Furthermore, LLP1 was necessary for Pin-induced immunity (Fig. 3b) and the restriction of *Pst* titers after exposure of the plants to the volatile emissions of SAR-induced wt plants (Fig. 5g). Similarly, G3P was dispensable for Pin-induced immunity[20] and the G3P-deficient *gly1-3* mutant was observed to be resistant to *Pst* after exposure to emissions from SAR-induced wt plants (Fig. 5h). In contrast, although *ald1* mutants were unable to mount Pin-induced defense against *Pst* (Fig. 3b), *ald1* plants responded normally with reduced *Pst* titers to the emissions of SAR-induced wt compared with mock-treated plants (Fig. 5f). These data suggest the presence of additional volatile cues in emissions of SAR-induced wt plants which are recognized in the absence of Pip. One of these might be MeSA[6,7]. To test this, we used *benzoic*

*acid/salicylic acid methyltransferase1* (*bsmt1*) mutant plants, which displayed strongly compromised MeSA emissions[33], while their α-pinene emissions after infection appeared somewhat reduced, but were not significantly different from those of infected wt plants (Supplementary Fig. 5). Notably, the resistance of wt receivers to *Pst* was enhanced in response to emissions of infected *bsmt1* plants (Fig. 5i). Reciprocally, MeSA accumulation was not required in receivers for enhanced resistance in response to emissions of infected wt plants (Fig. 5j). Thus, MeSA does not appear to act as a dominant defense cue in PTP volatile blends or in downstream defense signaling cascades. Alternatively, the *ald1* mutant might, for example, be more sensitive to differences in Pin concentrations or to the possible interplay of Pin with other bio-active VOCs (e.g., nonanal[34]) in the PTP volatile blend[20]. In support of this hypothesis, *ald1* receivers responded with enhanced resistance to the emissions of infected wt, but not to those of infected *azi1-2* mutant plants (Supplementary Fig. 6), whose emissions were perceived by wt receivers (Fig. 5c). Although we cannot exclude qualitative differences in the VOC emissions of infected wt and *azi1-2* mutant plants that remained undetected due to technical limitations, we detected reduced α-pinene and camphene emissions in *azi1-2* compared with wt plants (Fig. 4 and Supplementary Fig. 4). Perhaps these lower emissions were sufficient to be perceived by wt, but not by Pip-deficient *ald1* plants.

**AfTERPENE SYNTHASE24 produces volatile defense cues.** The *ggpps12* mutant has reduced monoterpene emissions due to a change in metabolite fluxes reducing the accumulation of the general monoterpene precursor geranyl pyrophosphate (GPP)[20,35]. Because we cannot exclude roles in SAR of other terpenes arising from these metabolite flux changes, we aimed to more firmly establish the role of monoterpenes in SAR. GPP is converted to different monoterpenes by specific TERPENE SYNTHASES (TPS) that in *A. thaliana* are encoded by a 32-member gene family[35]. In vitro, recombinant TPS24 protein

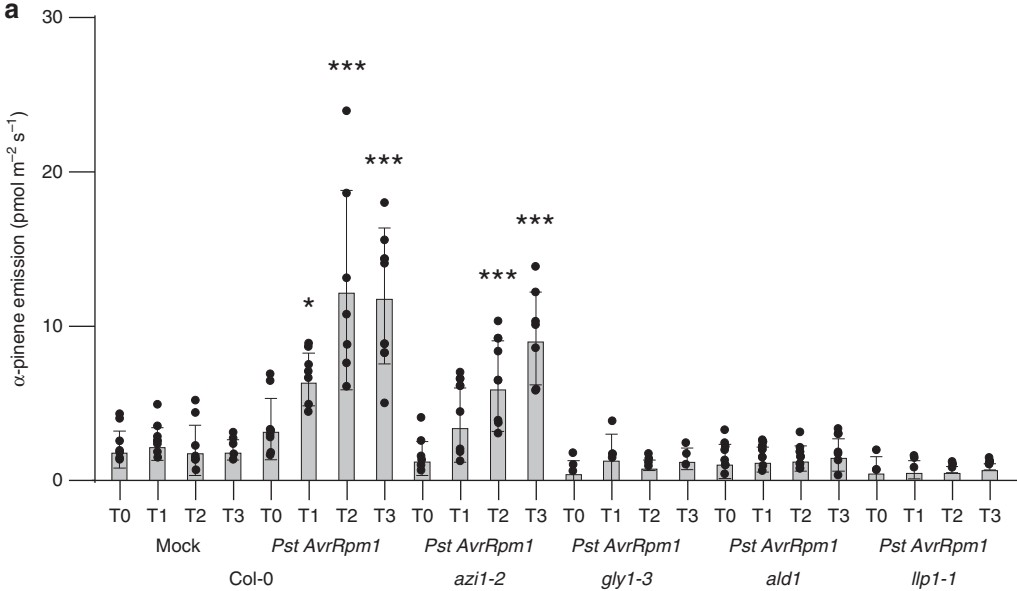

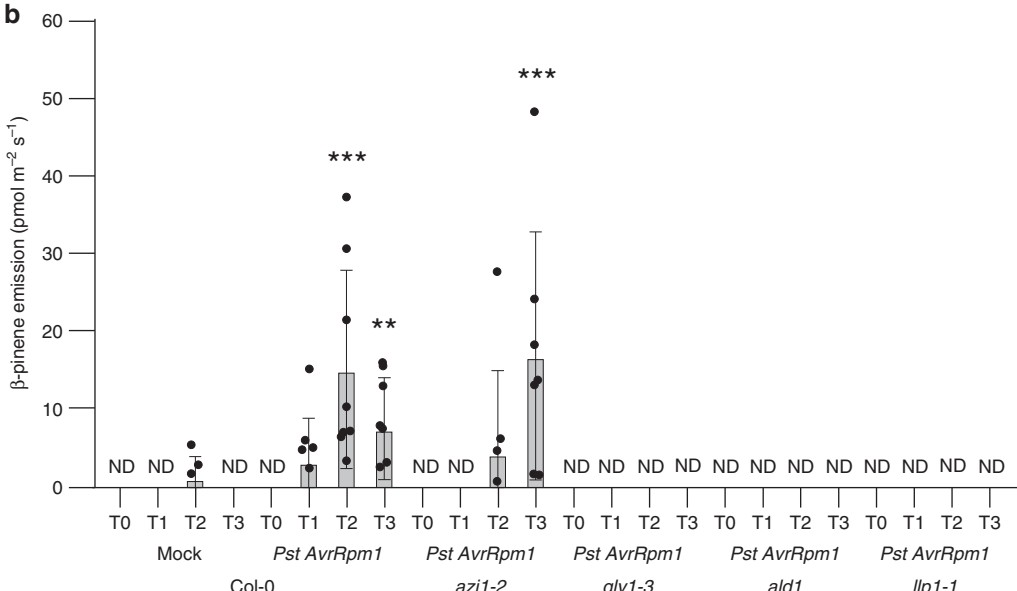

**Fig. 4** SAR-induced emissions of the volatile monoterpenes α- and β-pinene depend on LLP1, G3P, and Pip. Emission rates of α-pinene **a** and β-pinene **b** from the plant genotypes indicated below the panels 1 day before (T0) and during the first (T1), second (T2), and third (T3) day after spray inoculation of the plants with *Pst/AvrRpm1* or the corresponding mock treatment. Dots indicate individual, biologically independent results ($n = 3$–8); bars represent the average of the indicated results ± standard deviation. Asterisks indicate significant differences from the respective Col-0 mock control. Genotype main effect: Col-0\*\*\*, *azi1-2*\*\*\* (α-pinene); Col-0\*\*\*, *azi1-2*\*\*\* (β-pinene). Two-way ANOVA, multiple comparison versus control groups, Holm-Sidak method; \**P* < 0.05, \*\**P* < 0.01, \*\*\**P* < 0.001. ND: not detectable indicates that the average emission rates remained below background levels. Individual data that were below background levels are not indicated in the bars, but were included in statistical analyses of the data and in the source data file associated with this paper. Grubb's outlier test identified statistically significant outliers in the data sets Col-0 *Pst AvrRpm1* T1 and T2; these outliers were excluded from further analyses and are highlighted in gray in the source data file

when offered GPP as a substrate produces a monoterpene blend that includes α- and β-pinene[36]. Here, we used two independent *tps24* mutants carrying T-DNA insertions in the fourth intron and 3′ untranslated region of *TPS24* (Supplementary Fig. 7A). In spite of detectable *TPS24* transcript accumulation (Supplementary Fig. 7B), both *tps24* mutants had reduced α-pinene emissions after *Pst/AvrRpm1* infection compared with wt plants (Supplementary Fig. 5). In this experiment, all other monoterpenes remained below the limit of detection, also in emissions of wt plants. Therefore, we considered α-pinene as representative for monoterpenes and concluded from the reduced emissions in both

*tps24* mutants that *TPS24*, possibly together with other *TPS*, contributes to monoterpene emissions after infection. In support of an essential role of monoterpenes in SAR, both *tps24* mutants were unable to support SAR after a local *Pst/AvrRpm1* infection (Supplementary Fig. 8). Also, wt receiver plants did not mount enhanced resistance when exposed to the emissions of infected *tps24* mutants (Fig. 5i). Similar to *ggpps12* plants (Fig. 5b), both *tps24* mutants responded to the airborne cues from infected wt plants with enhanced resistance to *Pst* (Fig. 5j). These data strongly support a role of monoterpenes in SAR and as a volatile cue in PTP blends.

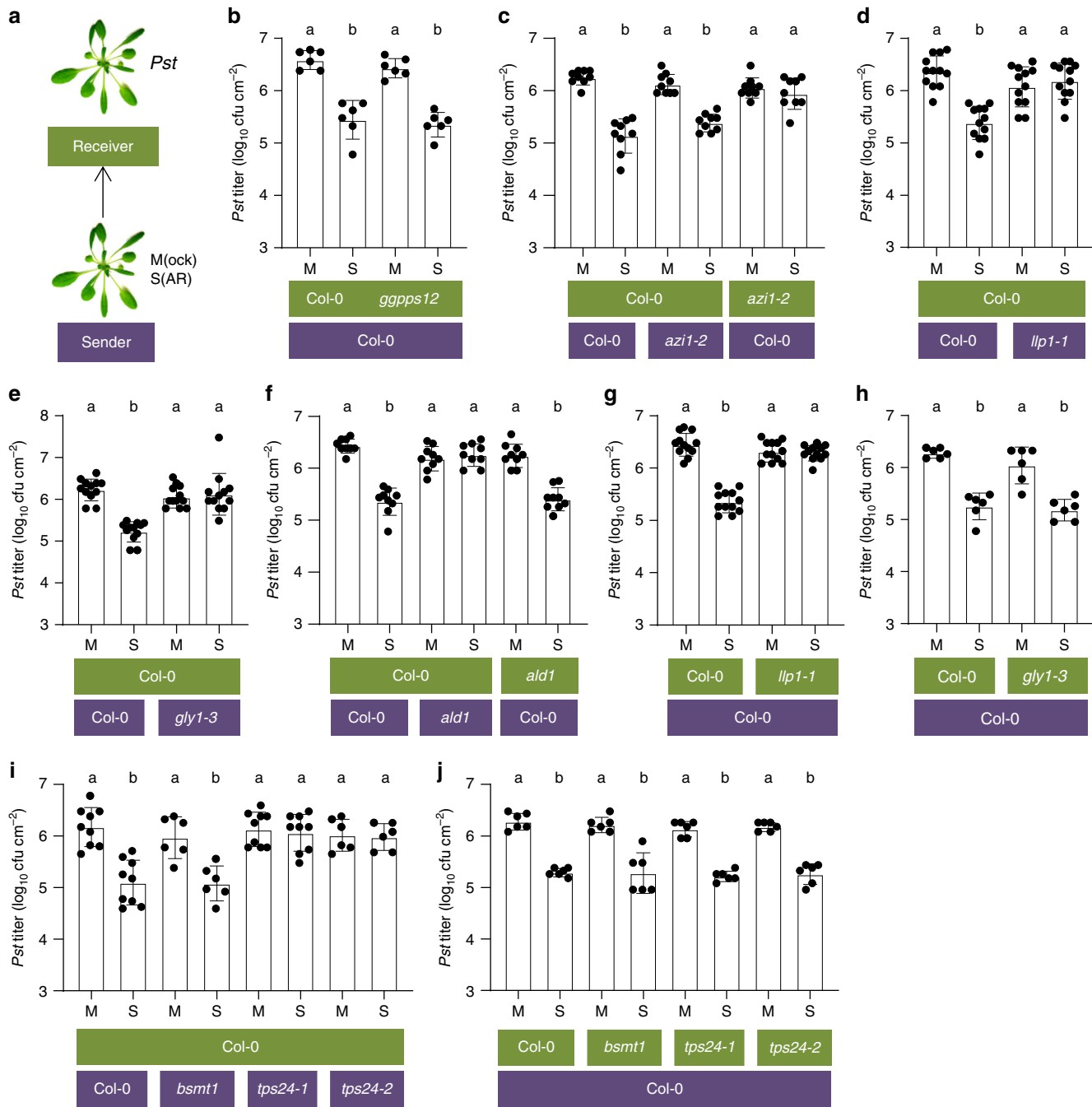

**Fig. 5** Plant-to-plant (PTP) propagation of innate immunity depends on LLP1, G3P, Pip, and *TERPENE SYNTHASE 24* (*TPS24*). **a** PTP experimental setup. Senders were either inoculated with *Pst/AvrRpm1* (SAR-induced; S) or mock-treated (M), and incubated with naive receivers in gas-tight containers. The receiver plants were subsequently released from the containers and inoculated with *Pst*. **b–j** In planta *Pst* titers at 4 dpi of the receiver plants. The plant genotypes are indicated below the panels (senders in purple and receivers in green) and include *bsmt1* with reduced MeSA emissions and *tps24* with compromised monoterpene emissions. The treatments of the senders (M or S) are indicated below the bars. Dots indicate individual results from two (**b**, **h**; *bsmt1* in **i**, **j**), three (**c**, **f**, **i** remaining genotypes), or more (**d**, **e**, **g**) biologically independent experiments, including three replicates each. Bars represent the average of the indicated results ± standard deviation. Different letters above bars indicate significant differences, one-way ANOVA, $P < 0.05$

**PTP immunity is subject to a self-fortifying feedback loop.**
Because LLP1 is necessary for both generation and recognition of PTP cues, we investigated if PTP immune signal transfer is subject to positive feedback regulation. To this end, we performed a normal PTP experiment as described above. Instead of inoculating the receiver plants, these primary receivers were used as secondary senders and a fresh set of naive plants was used as receiver 2 (Fig. 6a). In these experiments, receiver 2 plants supported reduced *Pst* growth if sender 1 had been SAR-induced as

compared with the response to a double mock treatment of both sender 1 and 2 (compare S- to MM in Fig. 6b). As a control, we mock-treated secondary senders that had been exposed to SAR-induced primary senders. Again, the immunity of wt receiver 2 plants to *Pst* was enhanced after their exposure to mock-treated secondary senders if sender 1 had been SAR-induced (compare SM with MM in Fig. 6b). These data suggest that plants perceiving PTP signals pass along the chemical cue to distant neighbors. Interestingly, this is dependent on the capacity of the

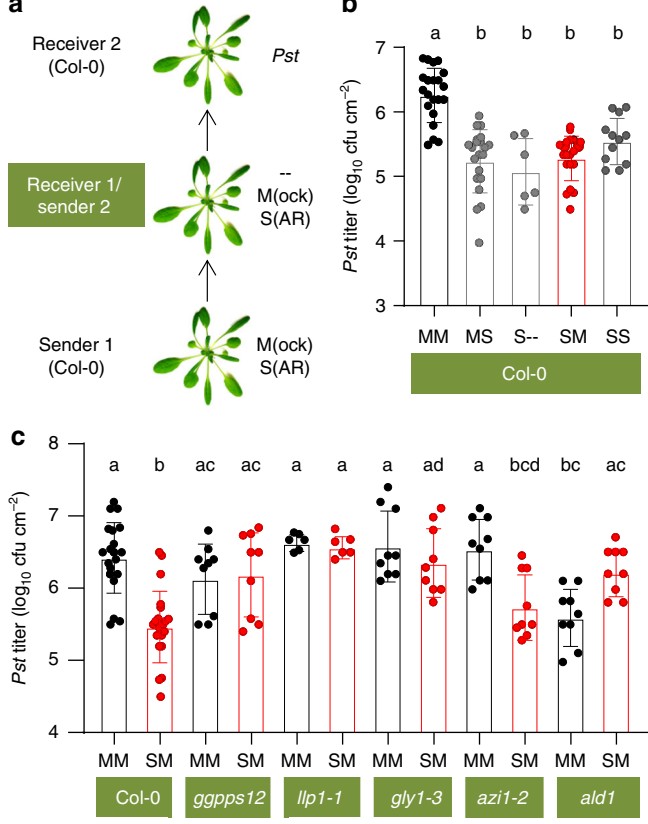

**Fig. 6** PTP innate immune cues are propagated in the absence of infection and depend on monoterpene emissions and the SAR signaling network. **a** Plant-to-plant-to-plant (PTPTP) experimental setup. Col-0 senders (sender 1) were SAR-induced (S) or mock-treated (M) and incubated with receiver 1 plants. These plants were either not further treated (–), mock-treated (M) or SAR-induced (S) and incubated as secondary senders (sender 2) with Col-0 wild-type secondary receivers (receiver 2). **b**, **c** *In planta* *Pst* titers at 4 dpi of receiver 2 plants. Sender 1 and receiver 2 were Col-0 wild type in all experiments; the genotypes of the receiver 1/sender 2 plants are indicated in green below the panels. Capital letters below the bars indicate the consecutive treatments of sender 1 and sender 2. Dots indicate individual results from two (S/– in (**b**) and *llp1-1* in (**c**)) to three or more biologically independent experiments, including three replicates each. Bars represent the average of the indicated results ± standard deviation. Different letters above bars indicate significant differences, one-way ANOVA, *P* < 0.05

middle plant to produce monoterpenes. Although *ggpps12* responded with enhanced immunity to PTP signals from SAR-induced wt plants (Fig. 5b), the same mutant did not pass along the PTP cue to wt receiver 2 plants (Fig. 6c), suggesting that propagation of the PTP cue is mediated by an active process that is associated with monoterpenoid biosynthesis.

We next compared the immunity of plants directly responding to PTP signals to that in plants exposed to secondary senders. *Pst* titers were comparable in plants directly exposed to PTP cues (MS in Fig. 6b) and in receiver 2 plants exposed to mock- or untreated sender 2 plants if sender 1 had been SAR-induced (SM and S- in Fig. 6b). This suggests that propagation of the PTP cue was stable with little if any loss of strength during transmission from sender 1 to sender 2. Adding an additional exposure round, in which a third set of plants (receiver 3) was exposed to the emissions of receiver 2 plants, likewise did not significantly alter the *Pst* titer reduction if sender 1 was SAR-induced (Supplementary Fig. 9). Vice versa, infection of both sender 1 and 2 plants with *Pst/AvrRpm1* did not further enhance resistance in receiver 2

plants (compare SS to SM in Fig. 6b). This suggests that superimposing SAR signaling induced by *Pst/AvrRpm1* onto PTP cue propagation did not significantly affect the defense-inducing capacity of the PTP volatile blend.

The *llp1-1* mutant neither generated nor responded to PTP cues (Fig. 5d, g). As expected, PTP cues also were not propagated in *llp1-1* secondary senders exposed to SAR-induced wt primary senders (Fig. 6c). Similarly, *gly1-3* and *ald1* mutant plants, which did not emit defense-inducing PTP cues (Fig. 5e, f) and exhibited reduced monoterpene emissions after *Pst/AvrRpm1* infection (Fig. 4 and Supplementary Fig. 4), did not propagate PTP cues after exposure to emissions of SAR-induced wt sender 1 plants (Fig. 6c). Together, the data suggest that LLP1, G3P, and Pip cooperate in the regulation of the emission and propagation of defense-inducing PTP cues, and that this converges on the ability of plants to emit monoterpenes. In contrast, the *azi1-2* mutant emitted monoterpenes (Fig. 4 and Supplementary Fig. 4) and PTP cues (Fig. 5c) after infection, and this was associated with the propagation of PTP cues if *azi1-2* was used as the middle plant in plant-to-plant-to-plant (PTPTP) experiments (Fig. 6c).

**Pip and G3P cooperate in PTP propagation of immunity.** The collective data presented so far suggest that SAR and defense cue accumulation in PTP volatile blends is regulated by a positive feedback loop involving Pip, G3P, LLP1, and monoterpenes. Because monoterpenes are centrally important in particular in PTP transfer of immunity, it appeared conceivable that monoterpenes could suffice to trigger the positive feedback loop. To test this hypothesis, wt plants were exposed to volatile Pin for 3 days and subsequently used as senders in a PTP experiment (i.e., Pin-to-plant-to-plant experiments; Fig. 7a). Wt receivers exposed to the emissions of Pin-treated plants mounted a resistance response to *Pst* compared with plants exposed to the emissions of mock-treated plants (Fig. 7b), suggesting that monoterpenes are necessary and sufficient to trigger PTP propagation of defense.

As part of the proposed LLP1-dependent SAR and PTP positive feedback loop, Pip might alone trigger PTP transfer of immunity. To test this, wt receiver plants were exposed to the emissions of plants that were irrigated with Pip (Fig. 7a). Similar to that in receivers exposed to the emissions of Pin-treated plants, the resistance of receivers to *Pst* was elevated in response to the emissions of Pip-treated wt plants (Fig. 7c). The SAR-defective phenotype of the Pip-deficient *ald1* mutant can be fully complemented by treating *ald1* plants with Pip prior to or during a primary SAR-inducing infection[11]. In contrast, *Pst* growth was not restricted in wt receivers exposed to the emissions of Pip-treated *ald1* plants (Fig. 7c), suggesting that this requires a SAR-inducing infection-associated co-factor that is present in wt but not in uninfected *ald1* plants. Because both Pip and G3P were necessary for monoterpene biosynthesis as well as emission and propagation of defense-inducing PTP cues (Figs. 4–6) and because G3P levels were lower in SAR-induced *ald1* mutant compared with wt plants[12], we hypothesized that this co-factor might be G3P. To test this, *ald1* mutant plants were treated near the roots with pip (or an $H_2O$ mock control) and the fully expanded leaves were subsequently infiltrated with G3P (or 10 mM $MgCl_2$ as a mock control). These treated plants were used as senders in PTP experiments. Similar to emissions from Pip-treated *ald1* plants, exposure of wt plants to the emissions from G3P-treated *ald1* plants did not enhance their resistance to *Pst* (Fig. 7d). However, perception of volatile emissions from *ald1* mutants treated with both Pip and G3P enhanced the resistance of receivers to *Pst* (Fig. 7d), suggesting that Pip and G3P together sufficed to trigger the emission of defense-inducing PTP cues.

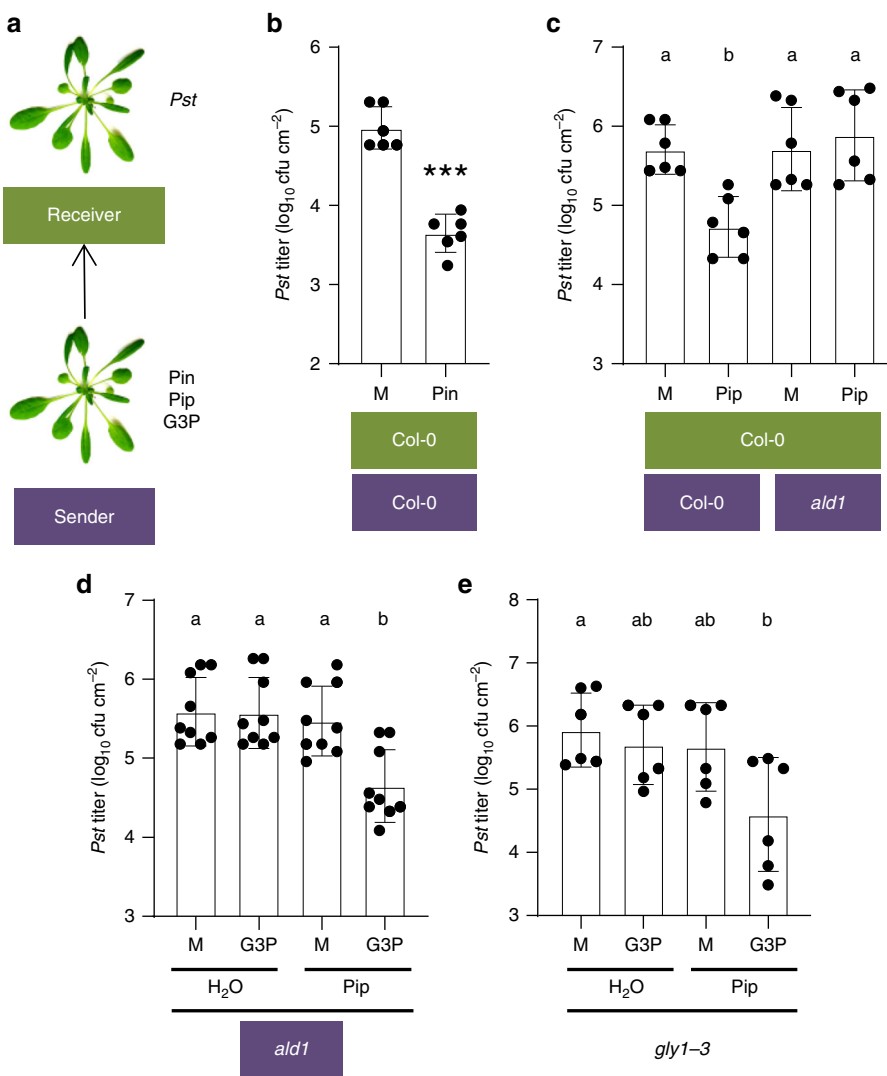

**Fig. 7** Pip and G3P cooperate to promote SAR and PTP cues. **a** Experimental setup of (**b**–**d**). Senders were exposed to Pin, treated with Pip and/or G3P, or exposed to the appropriate mock controls, and incubated with naive receivers in gas-tight containers. The receivers were subsequently released from the containers and inoculated with *Pst*. **b**, **c** *In planta Pst* titers at 4 dpi of the receivers. The plant genotypes are indicated below the panels (senders in purple and receivers in green) and the treatments of the senders are indicated below the bars. **d** Chemical complementation of the PTP-signaling defect of the *ald1* mutant with Pip and G3P. *ald1* senders were irrigated with Pip or the corresponding $H_2O$ control and subsequently syringe-infiltrated in the leaves with G3P or the corresponding mock control as indicated below the panel. The treated plants were incubated with Col-0 wt receivers in gas-tight containers. The receiver plants were subsequently released from the containers and inoculated with *Pst*. The resulting *in planta Pst* titers at 4 dpi are shown. **e** Chemical complementation of the SAR-deficient phenotype of the *gly1-3* mutant with Pip and G3P. *gly1-3* mutant plants were irrigated with Pip or the corresponding $H_2O$ control and subsequently syringe-infiltrated in the first two true leaves with G3P or the corresponding mock control as indicated below the panel. Leaves systemic to the site of G3P application were inoculated with *Pst*, and the resulting *in planta Pst* titers at 4 dpi are shown. **b**–**e** Dots indicate individual results from two (**b**, **c**, **e**) to three (**d**) biologically independent experiments, including three replicates each. Bars represent the average of the indicated results ± standard deviation. **b** Asterisks above bar indicate a significant difference from the mock control (*t* test, $P < 0.001$). **c**–**e** Different letters above bars indicate significant differences, one-way ANOVA, $P < 0.05$

The SAR-deficient phenotype of the G3P-deficient *gly1-3* mutant can be complemented by addition of G3P to the SAR-inducing primary inoculum[18]. However, G3P alone does not enhance systemic immunity and likely requires a co-factor that is present in petiole exudates of SAR-induced but also mock-treated wt *A. thaliana* plants[18]. Here, we tested if that co-factor might be Pip. To this end, *gly1-3* mutant plants were treated near the roots with Pip or $H_2O$ as a mock control. One day later, two lower leaves of the plants were infiltrated with G3P or 10 mM $MgCl_2$ as a mock control. Another 2 days later, systemic leaves of treated plants were inoculated with *Pst* and the resulting *in planta* titers monitored at 4 dpi. As expected, treatment of *gly1-3* with either

G3P or Pip alone did not elevate resistance to *Pst* compared with the respective mock treatments (Fig. 7e). However, Pip and G3P together sufficed to trigger systemic immunity to *Pst* in *gly1-3* plants (Fig. 7e). In conclusion, the data suggest that Pip and G3P together are necessary and sufficient to trigger within- and between-plant (systemic) immunity.

**PTP transfer of immunity is functional in open systems.** The PTP experiments described so far were performed in cuvette-enclosed static conditions, where due to limited air exchange, the build-up of VOCs inside the desicator might have risen to

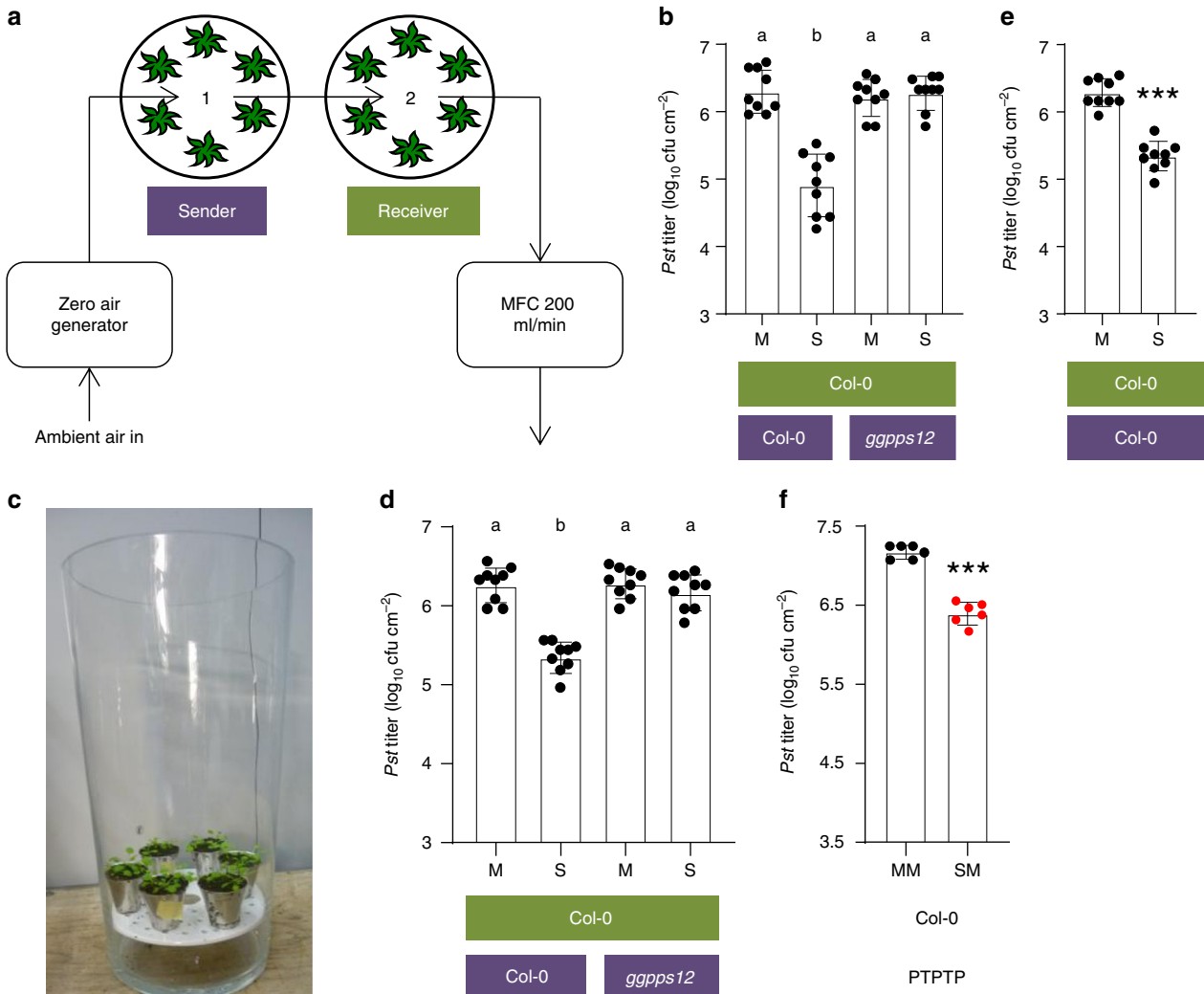

**Fig. 8** PTP propagation of innate immunity in open systems. **a** Experimental setup of PTP experiments under dynamic/flow-through conditions. Sender and receiver plants were incubated in separate, but via 1/8″ tubes connected vacuum desiccators. VOC-free air was derived from ambient air by a zero-air generator and pulled through the system at a flow rate of 200 mL min$^{-1}$. (**b**–**e**) PTP experiments in the dynamic system (**b**), open glass vases (**c**, **d**), and open desiccators (**e**). Receivers were exposed to the emissions of *Pst/AvrRpm1*-infected (SAR-induced; S) or mock-treated (M) sender plants in the different experimental setups (e.g., **a**, **c**). The plant genotypes are indicated below the panels (senders in purple and receivers in green); the treatment of the senders is indicated below the bars. After 3 days, receivers were inoculated with *Pst*, and the resulting *in planta* *Pst* titers at 4 dpi are shown. Dots indicate individual results from three biologically independent experiments, including three replicates each. Bars represent the average of the indicated results ± standard deviation. Different letters above bars indicate significant differences, one-way ANOVA, $P < 0.05$. **f** PTPTP experiment in Col-0 wt plants. Data summarize *Pst* titers in receiver 2 plants. Receiver 1 had been exposed to SAR-induced (S) or mock-treated (M) sender 1 plants in the flow-through system from (**a**) for 3 days. Subsequently, receiver 2 was exposed to mock-treated receiver 1 plants in the same setup. After another 3 days, receiver 2 plants were inoculated with *Pst*, and the resulting *in planta* *Pst* titers at 4 dpi are shown. Dots indicate individual results from two biologically independent experiments, including three replicates each. Bars represent the average of the indicated results ± standard deviation. Asterisks above bar indicate a significant difference from the mock control (*t* test, $P < 0.001$). MFC, mass flow controller

unrealistic concentrations. To ensure that PTP propagation of immunity also occurs in open systems, we additionally ran PTP experiments using flow-through dynamic cuvette systems or free-air conditions. To this end, sender and receiver plants were initially kept in separate vacuum dessicators that were connected and continuously flushed with 200 mL min$^{-1}$ of VOC-free air (Fig. 8a). Exposure of receiver plants to the emissions of *Pst/AvrRpm1*-infected plants reduced growth of a *Pst* challenge inoculum compared with that in receiver plants exposed to the emissions of mock-treated plants (Fig. 8b). The response of the receiver plants depended on the ability of the sender plants to generate monoterpenes, supporting our above conclusion that monoterpenes are essential defense cues in PTP blends. Similar

results were obtained when receiver plants were exposed to infected senders under free-air conditions using open-top vases (Fig. 8c, d) or open desiccators (Fig. 8e). Finally, PTPTP experiments were performed using the flow-through system (Fig. 8f). Primary and secondary exposures were conducted consecutively in order to prevent an inadvertent exposure of secondary receivers to emissions of primary sender plants. In these experiments, *in planta* titers of a *Pst* challenge inoculum were restricted in secondary receivers exposed to the emissions of primary receiver/secondary sender plants if the primary senders had been SAR-induced (Fig. 8f). Thus, PTP propagation of innate immunity via volatile cues is functional in flow-through or free-air conditions and dependent on monoterpenoid biosynthesis.

These data support a possible physiological relevance of this process, also in more natural settings.

## Discussion

In plant-insect interactions interorganismic interactions through volatiles are a well-known phenomenon regulating defense[21,23,37]. After insect attack, for example, plants emit volatile compounds that attract predators to reduce insect propagation[38]. Also, herbivore-induced plant volatiles (HIPVs) mediate defense propagation between plants and prime insect resistance in neighboring plants[22,23,39–42]. HIPVs also act within plants presumably complementing vascular long-distance signaling to reach tissues outside of the orthostichy[22,43–47]. Similarly, long-distance signaling in response to pathogen attack induces SA-mediated immunity or SAR in systemic plant parts outside of the orthostichy[48]. Since volatile pinenes are essential for within-plant SAR[20], volatile signaling between leaves of the same plant might well play a significant role in SAR-related long-distance signaling. We show here that SA immunity-inducing plant-to-plant cues are robustly propagated in receiver tissues and plants in a process that is mechanistically dependent on the within-plant SAR signaling network (Fig. 9). Our collective data suggest that within-plant SAR and plant-to-plant transfer of innate immunity depend on the same mechanism regulating (systemic) immunity at the individual and most likely also the population level.

LLP1 is a predicted lectin that acts systemically in the recognition of phloem-mobile SAR signals and volatile PTP cues (Figs. 1 and 5). Other known systemic components of SAR include cuticular waxes[49], Pip[12], and plasmodesmata-localizing proteins, possibly in interaction with AZI1[28]. Here, AZI1 was required for resistance establishment downstream of PTP cues in receivers (Fig. 5), and its role is most likely primarily confined to immune signal transduction farther downstream in the establishment of immunity. In addition to its role in SAR signal recognition, LLP1 together with LLP2 and LLP3 (and possibly additional similar proteins) promoted the generation or transmission of phloem-mobile signals (Fig. 1 and Supplementary Fig. 1). Similar to Pip and G3P, LLP1 was also essential for generation of innate immunity-inducing PTP cues (Fig. 4). Moreover, whereas Pip and G3P were not necessary for establishment of immunity downstream of PTP cues, both signals as well as LLP1, promoted propagation of PTP cues in primary receivers (Fig. 6). This suggests that Pip, G3P, and LLP1 are part of a positive feedback loop propagating PTP transfer of immunity and most likely also within-plant SAR signal generation (Fig. 9).

The *ggpps12* and *llp1* mutants did not respond to Pip irrigation with elevated resistance to *Pst* (Fig. 2), while *llp1* was equally unresponsive to Pin application and PTP cues (Figs. 3 and 5). These findings position Pip upstream of monoterpenes and LLP1 downstream of both in an initial defense-associated signaling cascade triggered after primary pathogen attack. Following activation of this pathway, LLP1 appears to drive a positive feedback loop to promote monoterpene-dependent propagation of PTP cues through Pip and G3P (Fig. 9). Exogenous G3P application alone could not rescue SAR in G3P-deficient *gly1* mutants[18] (Fig. 7). Similarly, wt receivers responded with enhanced resistance to PTP cues from Pip-treated wt plants, but not to those of Pip-treated *ald1* mutants (Fig. 7). Notably, co-application of G3P and Pip chemically complemented the SAR-deficient phenotype of the *gly1-3* mutant as well as the compromised emission of defense-inducing PTP cues from the *ald1* mutant (Fig. 7). Recent findings suggest that Pip and G3P accumulation is promoted by a SAR-associated positive feedback loop[12]. Thus, both compounds stimulate each other's accumulation and act together to trigger both within-plant SAR and PTP transfer of innate immunity (Fig. 9).

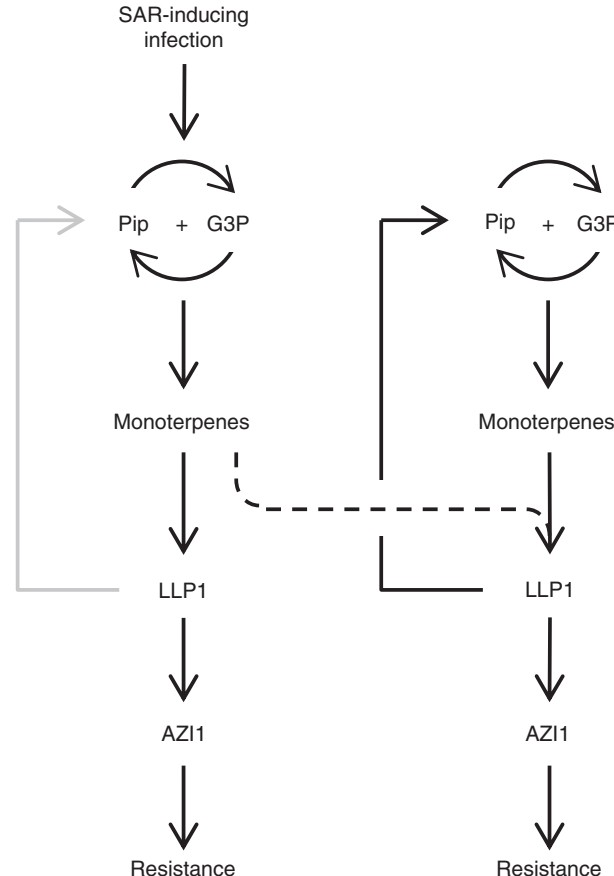

**Fig. 9** Working model of systemic and plant-to-plant propagation of innate immunity. A Systemic acquired resistance (SAR)-inducing infection triggers pipecolic acid (Pip) and glycerol-3-phosphate (G3P) accumulation, which stimulate each other in a positive feedback loop acting upstream of monoterpene emissions. Monoterpenes subsequently enhance salicylic acid (SA)-associated immunity through the SAR signaling intermediates LEGUME LECTIN-LIKE PROTEIN1 (LLP1) and AZELAIC ACID INDUCED1 (AZI1). At the same time, monoterpenes are emitted and act as cues that are perceived by systemic leaves and also neighboring plants. At the site of monoterpene perception, SA-associated immunity is enhanced through LLP1 and AZI1. In addition, LLP1 drives a positive feedback loop with Pip and G3P to stimulate monoterpene biosynthesis and emission, potentially promoting the generation of a wave of plant-derived volatile defense cues moving between leaves in a canopy or rosette and between neighboring plants in a population

The emission and propagation of PTP cues was associated with the ability of various SAR signaling mutants to emit monoterpenes (Figs. 4–6, Supplementary Figs. 4 and 5). This strongly suggests that monoterpenes are essential for between-plant transfer of innate immunity. Whereas we cannot fully exclude that other VOCs, which remained undetected by our methods, contributed to the response, such VOCs likely do not include MeSA (Fig. 5). In contrast, short pinene fumigation pulses of infected *ggpps12* plants were sufficient to complement the SAR-deficient phenotype of the mutant[20]. Moreover, fumigation of senders with α- and β-pinene triggered further propagation of PTP cues (Fig. 7), suggesting that these monoterpenes are essential in the odor profile and can suffice to induce PTP signaling. Hence, the data suggest that monoterpenes act as microbe-inducible plant volatiles (MIPVs), which as part of a plant-derived volatile mixture promote propagation of innate immunity in plant populations.

Similar to HIPVs[41,45], which prime jasmonic acid-associated defense responses against insects[22,23], MIPVs might contribute to both within-plant and between-plant propagation of SA-associated innate immunity, for example in dense natural or agricultural populations. In this working model, a single MIPV trigger might generate a wave of innate immune signals moving between closely positioned leaves or between neighboring plants leading to overall increased resistance within the canopy or plant stand. Such airborne cues to adjacent leaves and neighboring plants have been observed under natural or semi-natural conditions to influence plant susceptibility to pathogens or insect pests[22,42,44–46,50,51]. It can thus be envisioned that MIPV-triggered plant-to-plant cues allow development of future smart and sustainable agriculture systems that minimize the application of agrochemicals[52,53]. Self-fortifying volatile triggers, such as the monoterpenes presented here, should reduce the chemical burden of (crop) plant protection even more.

## Methods

**Plant material and growth conditions.** *A. thaliana* ecotype columbia-0 (Col-0) was used throughout this work. Mutants *llp1-1*, *gly1-3*, *azi1-2*, *ggpps12* (previously *ggr1-1*), and *ald1* were previously described[13,20,24,54,55]. The *tps24-1* and *tps24-2* mutants respectively correspond to T-DNA insertion lines SALK_127352 and SALK_055606, and were obtained from the Nottingham Arabidopsis Stock Center[56]. Seeds were propagated to homozygosity and homozygous mutants were used for all experiments. The T-DNA insertion sites were confirmed in two plants of each T-DNA insertion line. To this end, genomic DNA was isolated from leaves of 4-week-old plants and PCR products were generated using the primers pLBb1.3 and tps24-1-R or tps24-2-R, respectively (Supplementary Table 1; http://signal.salk.edu/tdnaprimers.2.html). The resulting fragments were sequenced (Eurofins Genomics, Germany) using pLBb1.3.

*RNAi:LLP1-3* constructs were generated by RT-PCR on RNA from untreated Col-0 plants. PCRs were performed with the primers LLP3-RNAi-F and LLP3-RNAi-R, with LLP2-RNAi-F and LLP2-RNAi-R, and with LLP1-RNAi-F and LLP1-RNAi-R (Supplementary Table 1). The resulting DNA fragments were annealed and amplified by PCR with primers LLP3-RNAi-F and LLP1-RNAi-R. The *RNAi:LLP1-3* target sequence was cloned into pENTR/dTOPO (Invitrogen), sequenced, and cloned into pHANNIBAL[57] in the sense and antisense orientations using *Xho*I/*Kpn*I and *Bam*HI/*Hind*III, respectively. The resulting RNAi cassettes were transferred to the binary vector pART27 using *Not*I. The resulting binary vectors were transformed into *Agrobacterium tumefaciens* strain GV3101 and used for plant transformation by floral dip[58]. Transgenic T1 plants were selected on Murashige and Skoog medium containing 50 µg mL−1 of kanamycin (Roth, Karlsruhe, Germany) and subsequently propagated to homozygosity. Experiments were performed in fourth or fifth generation (T4 or T5) homozygous *RNAi:LLP1-3* plants.

Plants were grown on normal potting soil mixed with silica sand (ratio 5:1) in 10-h days with a light intensity of 100 µmol m−2 s−1 photosynthetically active photon flux density at 22 °C and 14 h nights at 18 °C. Relative humidity (RH) was kept at ~70%. Four- to five-week-old plants were used for all experiments.

**Pathogens and infections.** *Pseudomonas syringae* pathovar *tomato* (*Pst*) and *Pst/AvrRpm1* were used for infections and SAR assays[24]. Bacteria were grown at 28 °C on NYGA medium (0.5% bacto proteose peptone, 0.3% yeast extract, 2% (v:v) glycerol, 1.8% agar, pH 7.0; Roth) containing 50 µg/mL kanamycin and 50 µg/mL rifampicin (Roth). For inoculation of plants, freshly grown bacteria were suspended in 10 mM MgCl₂; bacterial concentration of the suspension was determined using the formula $OD_{600} = 1.0$ equals $10^8$ colony forming units (cfu) per mL. To induce SAR, the first two true leaves of the plants were syringe-infiltrated with $10^6$ cfu per mL of *Pst/AvrRpm1* or with 10 mM MgCl₂ as the mock control. To induce transmission of SAR signals (for petiole exudate experiments), leaves 3 and 4 were syringe-infiltrated with $10^7$ colony forming units (cfu) per mL of *Pst/AvrRpm1* or with 10 mM MgCl₂ as the mock control. To analyze SAR, chemically induced resistance, or PTP-induced resistance, leaves 3 and 4 were syringe-infiltrated with $10^5$ cfu mL−1 of *Pst*. *In planta Pst* titers were determined 4 dpi[24]. To this end, bacteria were extracted by shaking (at 600 rotations per minute) 3 leaf discs per sample at room temperature in 10 mM MgCl containing 0.01% (v:v) Silwet. After 1 h, the samples were serially diluted in 10× increments; 20 µL per dilution were plated and grown on NYGA plates. After 2 days, bacterial titers in the leaves were calculated. Biologically independent replicates are defined as replicate experiments in independent plant batches performed at different times. Each replicate experiment includes at least three independently analyzed samples.

**Petiole exudate experiments.** Petiole exudate collection was modified from refs. [16,17]. Leaves were infected with *Pst/AvrRpm1* as described above and cut in the middle of the rosette 24 h later. The leaves were incubated with their petioles in 1 mM EDTA for 1 h and exudates were collected for 48 h in 2.0 mL of sterilized water using 6 leaves per exudate in the dark. The resulting petiole exudates were filter-sterilized (Millipore, 0.22 µm), supplemented with MgCl₂ to a final concentration of 1 mM, and syringe-infiltrated into two fully expanded leaves of naive recipient plants. Twenty-four hours later, the infiltrated leaves were either harvested for further analysis or inoculated with $10^5$ cfu mL−1 of *Pst*, *in planta* titers of which were determined at 4 dpi as described above.

**Chemical treatments.** MeSA, Pin, AzA, Pip, and G3P treatments were performed essentially as described[11,13,18,20]. MeSA (Sigma-Aldrich, Deisenhofen, Germany) and a mixture of α- and β-pinene (Sigma-Aldrich; molar ratio 1:1; Pin) were applied by fumigation[20]. Before each treatment, fresh MeSA and Pin solutions were prepared in hexane (Roth), and hexane was used for the mock control treatments in all fumigation experiments. For the treatments, 1.6 µmol of MeSA or 0.6 µmol of Pin was applied to filter paper and incubated in a vacuum dessicator with 12 plants for 3 days. Every 24 h, the air in the vacuum dessicators was exchanged and the treatment repeated. After 3 days, the plants were removed from the dessicators and infected with *Pst* as described above. AzA (Sigma-Aldrich) was dissolved in 50% MeOH (v:v) and diluted in 1 mM MgCl₂. One mM AzA in 0.025% MeOH or 0.025% MeOH (v:v) as the corresponding mock control treatment (each in 1 mM MgCl₂) was infiltrated into the first two true leaves of 4–5-week-old plants. Three days later, systemic leaves were inoculated with *Pst* as described above. Pip (DL-Pip; Sigma-Aldrich) was dissolved in H₂O at a concentration of 6.6 mM and applied to plant roots by irrigation with 3 mL per plant (~20 µmol of Pip per plant)[11]. H₂O was used as the corresponding mock control treatment. Two days later, two fully expanded leaves per plant were inoculated with *Pst*. G3P (Sigma-Aldrich) was dissolved in H₂O and diluted in 1 mM MgCl₂. Hundred µM G3P in 1 mM MgCl₂ or 1 mM MgCl₂ as the corresponding mock control treatment was syringe-infiltrated into the first two true leaves of 4–5-week-old plants. Three days later, systemic leaves were inoculated with *Pst*. If treatments were combined Pip irrigation was performed 1 day prior to G3P leaf infiltration. Systemic leaves were inoculated with *Pst* 2 days after the G3P treatment. For PTP experiments, Pin treatments were performed as described above and the treated plants were subsequently used as sender plants in PTP experiments as described below. In PTP experiments, Pip-irrigated and/or G3P-infiltrated sender plants were incubated with receiver plants immediately after the respective chemical treatments. G3P was infiltrated into 7 fully expanded leaves per sender plant. If treatments were combined, Pip irrigation was performed 1 day prior to G3P leaf infiltration. The treated plants were used as sender plants immediately after G3P infiltration.

**Plant-to-plant (PTP) and PTPTP experiments.** PTP experiments were performed essentially as described[20] in 5.5-L gas-tight glass desiccators (Rotilabo-Glas-Exsikkatoren, Roth) with plants grown in stainless steel pots containing 3–4 plants each. Sender plants were infected by spray inoculation with $10^8$ cfu mL−1 *Pst/AvrRpm1* in 0.01% Tween-20 (v:v) or with the corresponding 0.01% Tween-20 (v:v) mock control solution. Twelve *Pst/AvrRpm1*-infected or mock-treated senders were incubated with eight naive receivers in vacuum dessicators for 3 days. Every 24 h, the dessicators were opened for a short time to exchange the air. In PTP experiments, the receivers were infected with *Pst* as described above. In PTPTP experiments, 12 *Pst/AvrRpm1*-infected or mock-treated primary senders were incubated with 12 naive primary receivers for 3 days as above. The primary receivers were either left untreated, mock-inoculated, or infected with $10^8$ cfu mL−1 *Pst/AvrRpm1* by spray inoculation. Subsequently, these primary receivers were used as sender plants (secondary senders) and incubated with a fresh set of eight naive receivers (receiver 2 plants) in vacuum dessicators for 3 days. Every 24 h, the dessicators were opened to let in fresh air. After 3 days, the secondary receivers were infected with *Pst* as described above or used as senders in PTPTPTP experiments (in which case we used 12 secondary receivers) and incubated with another fresh set of naive receiver plants as described above.

PTP experiments in flow-through dynamic cuvette systems were performed using 18 *Pst/AvrRpm1*-infected or mock-treated senders in one vacuum desiccator and 12 receivers in a second vacuum desiccator. Each desiccator was equipped with air in- and outflow 1/8″ PTFA tubes with the outflow of the 'sender' desiccator connected to the inlet of the 'receiver' desiccator. VOC-free air, generated from ambient air using a ultra-high purity, organic free, zero-air generator (UHP-300ZA-S-E, Parker Hannifin Ltd, Tyne and Wear, UK), was pulled through the connected vacuum desiccators at a flow rate of 200 mL min−1. Receivers were exposed to the emissions of senders for 3 days and subsequently infiltrated in their third and fourth true leaves with *Pst*. Resulting *in planta Pst* titers were determined at 4 dpi. In PTPTP experiments the primary receivers were used as secondary senders during subsequent 3 days of exposure to secondary receiver plants in the same flow-through dynamic system consisting of two vacuum desiccators. In this manner exposure of secondary receivers to primary senders was avoided. Secondary receivers were subsequently challenged with *Pst* as described above.

PTP experiments in free-air conditions were performed in glass vases or open, glass (vacuum) desiccators as described above using 12 *Pst/AvrRpm1*-infected or mock-treated senders and eight receivers. Glass vases were 45 cm in height, and had the shape of a frustum with upper and lower base diameters of 23 and 19.5 cm, respectively.

**VOC analysis**. VOCs were collected essentially as described[20] from the headspace of 50 4.5-week-old plants, growing at temperature of 23.7 ± 0.7 °C, light conditions of 135 ± 15 μmol photons $m^{-2} s^{-1}$ photosynthetic active radiation (10-h photoperiod) and relative humidity of 50 ± 15%. Samples were taken the day before (T0) and during three consecutive days after the inoculation (T1–3). Plant VOC emissions were collected from three to eight biological replicates from at least two independent plant batches per treatment and genotype. Plants were either mock-treated or infected by spray inoculation with $10^8$ cfu $mL^{-1}$ Pst/AvrRpm1. Plants were enclosed in glass cuvettes (volume = ~4.2 L) flushed with 0.2 L $min^{-1}$ VOC-free synthetic air containing $CO_2$ at 400 ppmv. Air samples were collected in glass cartridges filled with 40 mg Tenax TA 60/80 and 40 mg Carbopack X 40/60 adsorbents (both from Sigma-Aldrich), and containing 250 pmol δ-2-carene as internal standard, at flow rates of 0.06 L $min^{-1}$ for 480 min for a total of 28.8 L volume. All air flows were controlled with calibrated mass flow controllers (MFC) (MKS Instruments GmbH, München, Germany). Each GC-MS cartridge containing a VOC sample was dried with ultra-pure $N_2$ (5.0) at a flow of 80 mL $min^{-1}$ for 1 h to remove moisture before analysis.

The VOC samples were analyzed with a thermo-desorption unit (TDU; Gerstel GmbH, Mülheim, Germany) coupled to a GC-MS instrument (GC-type, 7890A; MS-type, 5975C; Agilent Technologies, Waldbronn, Germany), using a 5% phenyl 95% dimethyl arylene siloxane capillary column (60 m × 250 μm × 0.25 μm DB-5MS + 10m DG, Agilent Technologies)[20]. The TDU-GC-MS was run as follows: samples were thermally desorbed by increasing the temperature from 35 to 270 °C at a rate of 280 °C $min^{-1}$, then cryo-refocused on Tenax TA at −50 °C for 0.31 min, and reinjected by ramping the temperature to 270 °C at a rate of 12 °C $min^{-1}$ and holding for 2 min. At the beginning of the run, inlet pressure was 15 psi. The GC-MS temperature program started at 40 °C followed by ramping at 10 °C $min^{-1}$ to 130 °C and holding for 5 min, then ramping at 80 °C $min^{-1}$ to 175 °C and holding for 0 min, then ramping at 2 °C $min^{-1}$ to 200 °C and holding for 0 min, then ramping at 4 °C $min^{-1}$ to 220 °C and holding for 0 min, then ramping at 100 °C $min^{-1}$ to 300 °C and holding for 6 min.

Mass spectra were acquired in both single ion mode (SIM) and SCAN mode (35–300 amu). VOCs were quantified in SIM mode and identified in SCAN mode. For monoterpenes, the ions 121 $m/z$ (dwell time, $d = 50$ ms) and 136 $m/z$ ($d = 50$ ms) were used. Temperatures of the MS source and quadrupole were 230 and 150 °C, respectively. Quantification was achieved by performing a calibration curve composed of six different concentrations of pure standards dissolved in hexane ranging between 20 and 250 pmol $μL^{-1}$. Each standard concentration was made independently in triplicate, and for each concentration also measured in triplicate.

The analyses included the measurements of 16 background replicates per time point, 8 taken before and 8 after the experiments. For background measurements, plants were removed from the soil before enclosing the pots inside the cuvettes. Because no significant changes were observed between the pre- and post-background measurements, their averages were used for background subtraction. The limits of detection (LOD) were set to 2*σ of background measurements. Emission rates (pmol $m^{-2} s^{-1}$) were based on projected leaf area (la). La was estimated from dry weight (dw) measurements using the conversion factor of 26.6 g $m^{-2}$ (dw $la^{-2}$) as determined in prior experiments under identical experimental conditions[20].

**Pip analysis (LC-MS)**. Plants were inoculated in their 3rd and 4th true leaves with $10^6$ cfu $mL^{-1}$ of Pst/AvrRpm1 in 10 mM $MgCl_2$ or with the corresponding 10 mM $MgCl_2$ mock control by syringe infiltration. Three days later, 100 mg of inoculated tissue was harvested, ground in liquid $N_2$, and resuspended in 1 mL 50% MeOH. Samples were shaken for 1 h at 1000 rpm and 4 °C and subsequently centrifuged at 13,000 rpm for 10 min. Hundred microliters of the supernatants were freeze-dried, and the dry matter dissolved in 100 μl 1:1 acetonitrile:$H_2O$ (v:v). The samples were centrifuged for 5 min at 14,000 rpm and 4 °C, and the supernatants were analyzed by Ultra Performance Liquid Chromatography Ultra-High Resolution tandem quadrupole/Time-Of-Flight mass spectrometry on a Ultimate 3000RS (Thermo Fisher, Waltham, MA, USA) coupled to Impact II with Apollo II ESI source (Bruker, Billerica, MA USA). Chromatography was performed on a BEH $C_{18}$ reverse-phase column (150 × 2.1 mm, 1.7 μm particles, Waters Technologies, Milford, MA, USA) with 0.2% of formic acid in water (eluent A) and acetonitrile 100% (eluent B). The gradient elution was as follows: hold of 5% B for 5 min, followed by an increase to 20% B until 7 min, 50% B until 8 min, 95% B until 9 min, decreasing to 70% B until 11 min, 50% B until 12 min, and 5% B until 14 min. The flow rate was set to 200 μL $min^{-1}$ and the column temperature maintained at 30 °C. The auto-sampler temperature was 8 °C. The MS was operated as follows: the nebulizer pressure was set to 2 bar, dry gas flow was 10 L $min^{-1}$, dry gas temperature was 220 °C, capillary voltage was 4000 V in positive mode, and the end plate offset was 500 V. Mass spectra were acquired in a mass range of 50–1300 $m/z$. Pip was identified using an authentic standard (Sigma-Aldrich) (retention time 2.1–2.4 min; $m/z$ 130.0860)[59] and quantified against an external standard curve with eight calibration points (ranging from 100 fg $μl^{-1}$ to 500 pg $μl^{-1}$, $R = 0.999$).

**RNA isolation and qRT-PCR**. Total RNA was isolated with Tri-Reagent (Sigma-Aldrich) according to the manufacturer's instructions. cDNA was generated with SuperscriptII reverse transcriptase (Invitrogen). qPCR was performed with primers from ref. [24] and Supplementary Table 1 using the Sensimix SYBR low-rox kit

(Bioline) and a 7500 real-time PCR system (Applied Biosystems). Transcript accumulation of target genes was analyzed using Relative Quantification with the 7500 Fast System Software 1.3.1. and normalized to that of the reference gene *UBIQUITIN*.

**Statistics**. GC-MS data presented in Fig. 4 and Supplementary Fig. 4 were evaluated as indicated in the respective figure legends using Sigmaplot version 11. qPCR, LC-MS (Pip), GC-MS data presented in Supplementary Fig. 5, as well as all Pst titer data were evaluated using Graphpad Prism Version 8 for Windows (version 8.1.1). Outliers were excluded using the Grubb's outlier test with α = 0.05. To ensure normal distribution of the data qPCR data were $log_2$-transformed and Pst titer data were $log_{10}$-transformed. Normal distribution of all data sets was verified using the Shapiro-Wilk test with α = 0.01. Subsequently, the data were analyzed using one-way ANOVA analyses with Tukey's multiple testing correction. Differences were considered significant with α = 0.05.

**Reporting summary**. Further information on research design is available in the Nature Research Reporting Summary linked to this article.

## Data availability
The data sets generated during and/or analyzed during the current study are available from the source data file associated with this paper.

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

## Acknowledgements

We thank Jean Greenberg for *ald1* seeds, Jürgen Zeier for *bsmt1* seeds, and Florian Hug for technical support. This work was funded in part by the Deutsche Forschungsgemeinschaft (DFG) as part of collaborative research centre (SFB)924 (project B06 to A.C.V.).

## Author contributions

M.W. conceived, designed, and performed experiments and analyzed the data; A.G. designed and performed experiments, analyzed the data, and edited the paper; J.H.S. and E.S.P. performed experiments and analyzed the data; H.H.B. and F.A. designed and performed experiments; B.W., B.L., and M.L. performed experiments and analyzed the data; R.K.C. and J.P.S. conceived and designed experiments and edited the paper; A.C.V. conceived and designed experiments, analyzed data, and wrote the paper.
