## [Peer Review File · Nature Communications]

Reviewers' comments:

Reviewer #1 (Remarks to the Author):

In this paper, the authors conduct a series of experiments with different SAR mutants to first establish the interaction between different SAR players (including potential systemic SAR signals and proteins required for SAR) in promoting systemic resistance. In a second series of experiments, the authors then evaluate the role of volatile pinenes in promoting SAR in neighboring plants and in transmitting SAR across multiple plants. As a result, the authors propose a model in which the different elements form a signaling cascade and potential positive feedback loop within which pinenes fortify the transmission of SAR from plant to plant.

Overall, this is an impressively detailed paper and an exercise in logic that provides detailed insights into the interplay of different SAR elements in protecting *A. thaliana* plants against Pst. Furthermore, the paper proposes an intriguing hypothesis, namely that induced pinenes may trigger a wave of SAR across a plant canopy, leading to canopy wide resistance rapidly upon infection of a few plants. I found this a very interesting and timely paper that has the potential to make an important contribution to the field of plant sciences, if some remaining weaknesses can be addressed.

Major comments:

This is a very thorough and interesting study, with one small, but important gap: The authors do not present any volatile measurements showing the release of Pin from sender plants. This is especially important for the PTP propagation experiments, where the middle plants should release Pin in response to Pin in order to propagate SAR in the canopy. This data would complete the manuscript. Conversely, without it, the reader remains to be convinced that Pin could propagate SAR across a canopy.

I would also feel more comfortable with this paper if the authors could show that the propagation of the volatile SAR cue also occurs in more natural conditions, i.e. in ventilated environments where volatiles do not build up over 24h in an airtight container. Systems like this have led to artifacts in the past, and it would be a pity to spend decades working on a phenomenon that turns out to be irrelevant in a canopy situation.

Please revise the use of language when it comes to use terms like "communication" and "signals". In organismic interactions, these terms are reserved for situations in which the emitter benefits from

releasing a cue by triggering a change in the other organism. Whenever this is unclear (as in this study), it would be more appropriate to talk about volatile cues instead of signals and take the perspective of the perceiving plant (i.e. talk about perception or eavesdropping instead of communication).

I am aware that this is a molecular biology paper, where statistics are often not very important, but it seems nevertheless that the statistical analyses are underdeveloped and somewhat problematic. Student's t-tests are normally OK for the comparisons at hand (although better models would be available to test for treatment x genotype interactions, such as two-way ANOVAs), but at the very least, the authors should verify that their data meets the assumptions of normality and equal variance (which seems to be violated in multiple cases, looking at the graphs), and adjust their statistics accordingly, either by data transformation or the use of robust statistical methods.

Further comments:

Page 2, line 2: This is not necessarily a form of communication. Revise language throughout the manuscript. I would recommend to avoid the term for plant-plant phenomena.

Page 2, line 4: Same for signaling- insufficient evidence that these are "signals" that are sent from plant to plant. Revise language according to communication theory.

Page 2, line 7: Can plant-plant transfer be called "systemic"? This term is typically reserved to within-plant transfer.

Page 2, line 12: triggers

Page 3, line 11: It would be good to discuss the current view on the importance of MeSA as a volatile systemic signal in different plants in order to put this work into context.

Page 3, line 15: Basal and systemic? Local and systemic or basal and induced, it would seem.

Page 3, line 16: Remove "the".

Page 3, line 29: I am not sure whether AZI1 should be considered to contribute to the systemic recognition of SAR signals. It would seem more logical to say that it is involved in the transduction of SAR signals?

Page 3, line 31: This statement is certainly not a reflection of the general state of the field. Volatile plant-plant signaling as a phenomenon in plant-herbivore interactions is not directly related to the idea of a "cry for help", and only a few people refer to the second phenomenon (the recruitment of natural enemies) as the "cry-for-help hypothesis".

Page 4, lines 16-19: Repetition from page 3.

Page 4, line 20: It would be helpful to highlight the open questions and hypotheses that form the basis of the present study, rather than just referring to an extension of the current knowledge on SAR.

Page 4, line 33: The ecological relevance sounds interesting, but is speculative at this point.

Page 7, line 32: Would this suggest that Pin is not required for SAR in receiver plants at all?

Page 9, line 10: Do the middle plants emit Pin upon exposure to the Pst-inoculated sender plants?

Reviewer #2 (Remarks to the Author):

This manuscript focuses the contributions of several important players to air-borne SAR. While this is an interesting topic, I think that this manuscript can be significantly improved by addressing the following concerns.

1. All the players in the current manuscript have been identified before. The current manuscript focuses on characterizing of existing players, instead of novel players.
2. The placement of upstream or downstream in the current manuscript could be relative depending on the type of treatments. Upstream components in a treatment could be downstream components in a different treatment.
3. In their natural environment, most plants grow in open space instead of containers or vacuum desiccators. It is not clear how the current study can be applied to major crops in the field.
4. Most of the data in the manuscript are descriptive. The underlying mechanisms remain elusive.
5. In terms of the difference of PR1 gene expression, most other papers use log₁₀ instead of log₂, indicating that the difference in the Figure 1B is quite small.
6. The previous paper has identified MeSA as a volatile signal. It is not clear to me the relationship between MeSA and pin.
7. It seems that three genes LLP1, LEC1 and LEC2 are involved. I believe that the authors should really use triple knockout lines instead of RNAi lines. I do not think the authors can use RNAi:LEC1LEC2LLP1 because one can't verify that no other gene is silenced besides these three.
8. If LLP1 is required for the production of pin and is involved in air-borne resistance. Why in the Figure 1, the author used exudates instead of testing air borne resistance
9. It states in the materials and methods "1.6 μmole of MeSA or 0.6 μmole of Pin was applied to filter paper and incubated in a vacuum desiccator with 12 plants for 3 d". Can you detect this level of MeSA or Pin if Arabidopsis plants were grown in open space? How about application of

MeSA and pin in open space instead of vacuum dessicators? In the Figure 4, it seems that the emission rate of pin is in pico molar range instead of μ mole used in the assays. What about the concentration of MeSA?

Reviewer #3 (Remarks to the Author):

The manuscript by Wenig et al. follows a previous publication by the same group (Riedlmeier et al., 2017) to further determine the role of volatile organic compounds (VOCs), specifically monoterpenes, in systemic acquired resistance (SAR) in Arabidopsis. In this study, the authors use several mutants of different SAR signaling components and a GPP synthase mutant impaired in leaf monoterpene biosynthesis to place monoterpene volatiles in the SAR signaling cascade. Moreover, the authors investigate these signaling responses in plants exposed to SAR-induced VOC emissions. The authors also show in an interesting set of experiments that receiver plants function as VOC emitters without further infection and cause a SAR immune response in a second receiver. Monoterpenes appear to be involved in the propagation of the immune signal.

Overall, the study is presented in carefully planned and interpreted experiments and provides new and deeper insight into the integration of VOC signals in intra-plant and inter-plant immune signaling. What concerns me is that the authors somewhat over-interpret their results in terms of VOCs generating waves of immune signals in plant populations or within canopies. The plant-plant interaction experiments were conducted in closed desiccators, which allow accumulation of VOCs over a period of 24 h. While this experimental setup serves well for the purpose of studying VOCs as signals in SAR, it is uncertain whether a similar outcome would be observed in an open chamber or flow through system with lower VOC concentrations. The “boost” of volatile signals across multiple plants seems to me questionable in natural environments and the authors should revise their interpretation or show that the plant-plant interaction experiments are possible in an open flow system.

It appears from Figure 5F (ald1 response to VOCs from Col-0 WT) that other volatile compounds than monoterpenes are also involved in the immune response. I recommend that the authors test other possible WT-emitted VOC candidates for their role in SAR to address this question. I am also wondering to what extent VOC profiles vary between the different mutants. Could other VOCs than monoterpenes be affected in these mutants? Likewise, the ggpps12 mutant does not release monoterpenes; but, I would find a more thorough comparison of SAR-induced or plant sender-induced VOCs of this mutant with those of wild type plants helpful to exclude possible off-site effects or effects of the lack of a GPP pool on other terpene VOCs.

Response to the reviewers' comments:

We thank all reviewers for their constructive and detailed comments. We paste our responses after each specific comment of each reviewer.

Reviewer #1:

Major comments:

This is a very thorough and interesting study, with one small, but important gap: The authors do not present any volatile measurements showing the release of Pin from sender plants. This is especially important for the PTP propagation experiments, where the middle plants should release Pin in response to Pin in order to propagate SAR in the canopy. This data would complete the manuscript. Conversely, without it, the reader remains to be convinced that Pin could propagate SAR across a canopy.

Response: This is a very important point that we addressed in the following ways: The volatile emission data shown in Figure 4 are emissions from plants treated in exactly the same way as sender plants in PTP and sender 1 plants in PTPTP experiments. In this sense, emissions from sender plants are thus presented. Please be aware that *A. thaliana* is a low emitter and that even in these experiments all GC-MS peaks were integrated by hand and scrutinized against background emissions from soil.

Unfortunately, it was technically not possible to generate the necessary number of 'middle plants' on the necessary small soil surface area to measure monoterpenes in the emissions of these plants that were above background levels. Because we could thus not provide unequivocal evidence that monoterpenes, or more specifically pinenes, are sufficient as plant-to-plant cues in the propagation of defense between plants, we rephrased and concluded that monoterpenes, and in particular pinenes, contribute to PTP propagation of innate immunity as part of a plant-derived volatile blend (page 2, line 14 and page 16, line 25).

I would also feel more comfortable with this paper if the authors could show that the propagation of the volatile SAR cue also occurs in more natural conditions, i.e. in ventilated environments where volatiles do not build up over 24h in an airtight container. Systems like this have led to artifacts in the past, and it

would be a pity to spend decades working on a phenomenon that turns out to be irrelevant in a canopy situation.

Response: We fully agree that the potential physiological relevance of PTP signaling in more open/ventilated/dynamic systems is important. In the revised work, we therefore included experiments using vacuum desiccators that were connected with silicon tubing allowing a regulated airflow from desiccator 1, containing sender plants, to desiccator 2, containing receiver plants. Results presented in the new main Figure 8 and described on page 14, lines 1-22 show that receiver plants responded with enhanced resistance to *Pst* in response to PTP cues from infected sender plants. This response was dependent on the ability of the sender plants to emit monoterpenes and subject to propagation in PTPTP experiments exposing receiver 2 to sender 2 plants in the dynamic flow-through system (Figure 8). This data demonstrates that PTP propagation of innate immune signals between plants occurs in ventilated systems. Moreover, similar results were observed in free-air systems, including open-top glass vases or open (vacuum) desiccators (in Figure S9). Together, the data strongly support our hypothesis that PTP signaling could well be relevant in natural systems.

Please revise the use of language when it comes to use terms like "communication" and "signals". In organismic interactions, these terms are reserved for situations in which the emitter benefits from releasing a cue by triggering a change in the other organism. Whenever this is unclear (as in this study), it would be more appropriate to talk about volatile cues instead of signals and take the perspective of the perceiving plant (i.e. talk about perception or eavesdropping instead of communication).

Response: Thank you for pointing this out. We rephrased whenever possible to view PTP and PTPTP results from the perspective of the receivers. When referring to PTP or PTPTP data, the word signal was replaced by cue; when referring to SAR, we continued to use the term signal. The term communication was carefully avoided throughout the revised manuscript.

I am aware that this is a molecular biology paper, where statistics are often not very important, but it seems nevertheless that the statistical analyses are underdeveloped and somewhat problematic. Student's t-tests are normally OK for the comparisons at hand (although better models would be available to test for treatment x genotype interactions, such as two-way ANOVAs), but at the very least, the authors should verify that their data meets the assumptions of normality and equal variance (which

seems to be violated in multiple cases, looking at the graphs), and adjust their statistics accordingly, either by data transformation or the use of robust statistical methods.

Response: We thank the reviewer for their kind wording and took up this very fair point in an effort to improve the statistical analysis of the data. The revised figures (old and new) include all available data rather than those of representative experiments. The data were analyzed using GraphPad Prism. First, Shapiro-Wilk's tests were used to assess normal distribution of the data. In some cases, *Pst* titer data required \log_{10} transformation to assure normality, and therefore we consistently \log_{10} transformed all data sets containing *Pst* titers. qPCR data were \log_2 -transformed to assure normal distribution. If necessary, outliers were removed using a Grubb's outlier test. By merging data from multiple biologically independent replicate experiments, the need for two-way ANOVA analysis was resolved; previously noted possible differences in the ratio between different treatments and genotypes were no longer observed. Therefore, all *Pst* titer and qPCR data was tested using one-way ANOVA.

Further comments:

Page 2, line 2: This is not necessarily a form of communication. Revise language throughout the manuscript. I would recommend to avoid the term for plant-plant phenomena.

Response: Thank you, this was done as outlined above.

Page 2, line 4: Same for signaling- insufficient evidence that these are "signals" that are sent from plant to plant. Revise language according to communication theory.

Response: Thank you, this was done as outlined above.

Page 2, line 7: Can plant-plant transfer be called "systemic"? This term is typically reserved to within-plant transfer.

Response: We agree that this is probably a matter of definition and removed the word systemic from this sentence. In all other cases, the word systemic was put in brackets if referring to both within- and between-plant transfer.

Page 2, line 12: triggers

Response: Thank you, this was changed to cues.

Page 3, line 11: It would be good to discuss the current view on the importance of MeSA as a volatile systemic signal in different plants in order to put this work into context.

Response: Yes, we agree; the first paragraph of the revised Introduction was extended to contain a more detailed introduction of MeSA and its possible role in SAR (page 3, lines 11-16).

Page 3, line 15: Basal and systemic? Local and systemic or basal and induced, it would seem.

Response: This part of the sentence was deleted.

Page 3, line 16: Remove "the".

Response: This was done.

Page 3, line 29: I am not sure whether AZ11 should be considered to contribute to the systemic recognition of SAR signals. It would seem more logical to say that it is involved in the transduction of SAR signals?

Response: This sentence was lost from the Introduction on editing, but we implemented your suggestion in a similar sentence in the revised Discussion (page 15, lines 8-11).

Page 3, line 31: This statement is certainly not a reflection of the general state of the field. Volatile plant-plant signaling as a phenomenon in plant-herbivore interactions is not directly related to the idea of a "cry for help", and only a few people refer to the second phenomenon (the recruitment of natural enemies) as the "cry-for-help hypothesis".

Response: Thank you. This was edited out of the revised Introduction.

Page 4, lines 16-19: Repetition from page 3.

Response: Thank you, this was deleted.

Page 4, line 20: It would be helpful to highlight the open questions and hypotheses that form the basis of the present study, rather than just referring to an extension of the current knowledge on SAR.

Response: Thank you. The final paragraph of the Introduction was significantly changed to highlight open questions in the SAR field, in particular with a view on systemic signal transfer (page 4, line 23 to page 5, line 4).

Page 4, line 33: The ecological relevance sounds interesting, but is speculative at this point.

Response: We agree. Although we now provide additional evidence that PTP propagation of defense occurs in ventilated conditions, we toned this sentence down to 'suggest' a possible ecological relevance on page 5, line 3-4.

Page 7, line 32: Would this suggest that Pin is not required for SAR in receiver plants at all?

Response: No, not in the least. Pin needs to be perceived as a cue for the receiver plant to respond with enhanced resistance and with the emission of further defense-inducing PTP cues. However, Pin biosynthesis does indeed not appear to contribute to further downstream resistance responses. We have added this conclusion to the relevant paragraph of the revised Results section on page 8, lines 28-30.

Page 9, line 10: Does the middle plants emit Pin upon exposure to the Pst-inoculated sender plants?

Response: Please see our response to your first comment above.

Reviewer #2 (Remarks to the Author):

1. All the players in the current manuscript have been identified before. The current manuscript focuses on characterizing of existing players, instead of novel players.

Response: We agree with reviewer 2 that it is known - especially through our own previous work - that monoterpenes can play a role in the development of systemic acquired resistance (SAR). In this paper, however, we demonstrate that monoterpenes are now linked to a new biological process. Molecular actors that were previously specifically associated with SAR and phloem-associated long-distance signaling are now also linked to plant-to-plant reactions. In addition, we demonstrate that the spread of innate immunity from plant to plant is subject to a self-enhancing positive feedback loop supported by the same molecular actors. This represents a new enhanced function for this mechanism by supporting

the dissemination of warning cues in plant populations. Similar processes have been observed in interactions between plants and insects, but the underlying mechanisms have not yet been resolved. With the present work, we outline a complete mechanism and unravel new molecular details within the associated signaling cascade, in particular the interdependence of Pip and G3P, which together are necessary and sufficient to promote both systemic acquired resistance and the spread of innate immunity from plant to plant.

To address the concerns of all reviewers (especially Reviewer 3), we have investigated new *A. thaliana* mutants of the gene TERPENE SYNTHASE 24 (TPS24). In contrast to GGPPS12, which causes the emission of monoterpenes via its effect on the chloroplastic GPP pool, TPS24 is a classical monoterpene synthase. Thus, the mutant phenotypes of *tps24* are closer and more specifically related to reductions in monoterpene emissions than those of *ggpps12* mutants. In fact, in two independent T-DNA insertion lines of *tps24* we observed a reduced emission of alpha-pinene after bacterial infection. More importantly, both SAR and the emission of defense-inducing PTP cues are absent in these mutants, which illustrates the essential role of this enzyme (TPS24) in both processes. The new data characterizing the role of TPS24 as a new player in both SAR and PTP propagation of immunity are included in main Figure 5I/J and in Figures S5 and S7 and are described on page 10, lines 10-27.

2. The placement of upstream or downstream in the current manuscript could be relative depending on the type of treatments. Upstream components in a treatment could be downstream components in a different treatment.

Response: That is a good and very important remark. Nevertheless, it proved very difficult to avoid using the terms upstream and downstream. Particularly in the Results section, we believe that these words help to keep the text legible. We have tried to take this remark into account in the new version of our work and have significantly shortened the discussion, in particular to reduce the use of the terms upstream and downstream and to refer directly to the circularity that this mechanism seems to have (e.g. page 15, lines 27-31). When using the terms upstream and downstream, we clarify that we refer to primary signaling events after a primary, SAR-inducing infection (e.g. page 15, line 24) or to the establishment of (systemic) defense responses to that primary trigger (e.g. page 7, lines 22 and 34).

3. In their natural environment, most plants grow in open space instead of containers or vacuum dessicators. It is not clear how the current study can be applied to major crops in the field.

Response: Please note that you share this concern with reviewer 1 and reviewer 3 (see also our reply to their comments above and below). In new additional experiments, we have tried to overcome this concern by conducting PTP experiments with wildtype plants and ggpps12 mutants in (1) ventilated and (2) open conditions. (1) In the first approach, we cultivated sender and receiver plants in separate desiccators with the airspace connected by silicone tubes. VOC-free air was transferred from the sender plants to the receiver plants by a regulated, dynamic gas flow. This ensured that VOCs and water vapor could not accumulate in the headspace surrounding the receiver plants. The experiments clearly demonstrated that both PTP and PTPTP propagation of innate immunity were functional with this arrangement and that the recognition of volatile cues by the receiver plants depended on the ability of the sender plants to emit monoterpenes. (2) To better document the importance of monoterpenes for PTP propagation of immunity, we cultivated receiver plants in open glass vases or open desiccators together with sender plants. Even in these completely open systems, the receiver plants reacted to PTP cues with enhanced resistance. In fact, we believe that the resistance response of the receiver plants to infected sender plants was more robust under aerated or open conditions than in closed containers. We attribute this to the fact that there is no accumulation of water vapor and no change in CO₂ concentration in these systems. The new data are included in the new main Figure 8 and in Figure S9 and are described on page 14, lines 1-22. While we believe that experiments with crop plants go beyond the scope of this manuscript, we are confident that these new data clearly demonstrate the potential physiological relevance of PTP cues in the spread of innate immunity of plants in natural environments.

4. Most of the data in the manuscript are descriptive. The underlying mechanisms remain elusive.

Response: With all due respect to reviewer 2's expertise in the field of plant to plant communication, we respectfully disagree with this statement. As already mentioned, we not only document a new biological process with potential ecological relevance, but also provide a complete mechanism to support it. Although only recently discovered, this paper provides the best characterized molecular mechanism for communication between plants. It has taken many years of research in the field of systemic immunity to uncover signaling links between different actors, and our work unravels another piece of this puzzle in the form of the observed link to the two non-volatile long-range transport molecules Pip and G3P. We then go on to connect this mechanism to a new biological process with important implications for plant protection and the conservation of biodiversity in ecosystems.

5. In terms of the difference of PR1 gene expression, most other papers use log10 instead of log2, indicating that the difference in the Figure 1B is quite small.

Response: We agree with this reviewer. In fact, this was the reason for including the resistance induction experiment in Figure 1C, which demonstrated a >10-fold reduction of *Pst* titers in tissues that were infected after their treatment with petiole exudates. The relatively low induction of *PR1* gene expression thus appears to be sufficient for successful resistance to *Pst*.

6. The previous paper has identified MeSA as a volatile signal. It is not clear to me the relationship between MeSA and pin.

Response: We thank reviewer 2 for raising this point. Our previous work (Riedlmeier et al., 2017, Plant Cell) reported on a possible role of volatile pinenes in intra- and inter-plant communication. However, we could not rule out the possibility that methyl salicylate (MeSA) might be involved in these reactions. Therefore, for the revised version of this manuscript, we conducted supplementary experiments with benzoic and salicylic acid methyltransferase1 (*bsmt1*) mutants that are inhibited in their accumulation of MeSA. As shown in the new figure S5, this mutant emits a similar amount of pinenes as wildtype plants after infection. It is important to note that wildtype receiver plants responded with increased resistance to *Pst* to the remaining volatile emissions of infected *bsmt1* senders (as shown in the revised Figure 5I). In addition, *bsmt1* receiver plants reacted normally with increased resistance to *Pst* to the emissions of infected wild type senders (as shown in the revised Figure 5J). Thus, in contrast to monoterpenes, which are essential components of defense-inducing PTP cues, MeSA does not appear to play a causative role in PTP propagation of innate immunity, neither in the sender nor in the receiver plants.

Previous work by others (Shulaev et al., 1997, Nature; Koo et al., 2007, Plant Mol Biol) has shown that MeSA can be effective as an airborne defense signal in tobacco and *A. thaliana*. In *A. thaliana*, the volatile emissions of *BSMT1*-overexpressing lines contained defense signals detected by *A. thaliana* neighbors. However, it is not clear from these studies whether MeSA was the causative signal and whether naturally occurring concentrations are sufficient. Our work clearly shows the relevance of monoterpenes as PTP defense signals, while the loss of naturally occurring MeSA in *bsmt1* mutants did not negatively affect the response. We therefore consider it important to show these data in the revised main Figure 5I/J and in Figure S5 (described on page 9, lines 20-28) in order to avoid any doubt that the

observed PTP responses are triggered by monoterpenes and not by MeSA. We do not discuss the role of MeSA as a possible phloem-mobile signal in SAR.

7. It seems that three genes LLP1, LEC1 and LEC2 are involved. I believe that the authors should really use triple knockout lines instead of RNAi lines. I do not think the authors can use RNAi:LEC1LEC2LLP1 because one can't verify that no other gene is silenced besides these three.

Response: Please note that we have taken the liberty of (re)naming the genes we referred to as LEC1 and LEC2 in the previous version of this work. LEC1 was formerly published as LEC (Lyou et al., 2009, Mol Cells), but this name has not been stored in known databases and leads to confusion with LEAFY COTYLEDON, another locus associated with another biological process. We therefore propose the names LLP2 and LLP3 for these close homologs of LLP1.

We consider it important to show that LLP1, together with LLP2 and LLP3, plays a role in the generation of SAR signals. These data add value to the reduced *PR1* induction in wildtype plants responding to petiole exudates from infected *llp1* mutant plants presented in Figure 1B. Also, they provide support to our hypothesis that LLP1 contributes both to the generation and recognition of SAR signals as it does to PTP cue emission and recognition if the model presented in Figure 9, as we postulate, is valid for both processes.

Please note that mutation of LLP2 and LLP3 is not trivial. We have a single T-DNA insertion KO line of LLP3 and no other lines are available. For LLP2, no KO lines are available and we have been unable to produce viable miRNAi lines. We additionally tried to silence LLP2 and LLP3, and obtained only lines with reduced LLP3 transcript levels; the two triple silencers included in this work were the only lines with reduced LLP1, 2, and 3 transcript levels. In addition, we tried various methods to over express LLP1, 2, or 3, and obtained few if any over expressors and lost over expression in homozygous T3 plants. Taken together, these genes and their expression levels appear quite essential for *A. thaliana*, and we feel that phenotypes in these lines are of potential biological/mechanistic interest and should remain included. We have, however, added sentences to state that we cannot exclude co-silencing of additional similar genes on page 6, line 16 and page 15, line 13.

8. If LLP1 is required for the production of pin and is involved in air-borne resistance. Why in the Figure 1, the author used exudates instead of testing air borne resistance

Response: At the time when we performed these experiments, the role of LLP1 in airborne resistance was not yet clear to us. Although we are now aware of this role, we felt that it is important to show that LLP1 likewise plays a role in phloem-mobile resistance signaling. This allowed for further connections between LLP1 and other signals associated with phloem-mobile long distance signaling, and supported our working hypothesis that SAR and airborne resistance rely on the same molecular mechanism. Experiments characterizing the role of LLP1 in airborne resistance are included in main Figures 5 and 6.

9. It states in the materials and methods “1.6 μ mole of MeSA or 0.6 μ mole of Pin was 31 applied to filter paper and incubated in a vacuum dessicator with 12 plants for 3 d”. Can you detect this level of MeSA or Pin if Arabidopsis plants were grown in open space? How about application of MeSA and pin in open space instead of vacuum dessicators? In the Figure 4, it seems that the emission rate of pin is in pico molar range instead of μ mole used in the assays. What about the concentration of MeSA?

Response: In our experiments (Riedlmeier et al., 2017) we have used MeSA as a positive control with a concentration that we experimentally deduced as effective. Please note that we apply ~5-fold less MeSA than e.g. Park et al. (2007, Science) and comparable studies. Also, the detection of MeSA with our GC-MS method is not as sensitive as for monoterpenes. This is due to its poor signal-to-noise ratio compared to that of monoterpenes which leads to a higher Limit of Detection (LOD) for MeSA. In several rounds of experiments, MeSA was detectable only once above the background values and this was in the emissions of infected ald1 mutants. Nevertheless, we removed these data from the revised manuscript because we could not detect any MeSA in the emissions of wildtype plants and were concerned about possible artifacts.

Concerning the concentration of pinenes we applied on the filter papers: The concentration of pinenes we used on the filters was ~1000 times higher than what we have measured in plant emissions. However, in new experiments in Figure 8 and S9 we demonstrate that natural Arabidopsis emissions (under closed and open headspace conditions) are sufficient to trigger the response in the eavesdropping neighbors.

As also discussed in our previous work (Riedlmeier et al. 2017, Plant Cell) and in the current manuscript, we cannot (and do not want to) exclude additional so far undetected volatile signals in plant volatile mixtures that interact as airborne defense signals. Nevertheless, the present data show that PTP cues are emitted and recognized and that the biological response in receiver plants depends on monoterpene

biosynthesis in the sender. Therefore, we believe that the data convincingly demonstrate that monoterpenes, especially pinenes, play an essential role in a defense-inducing plant volatile mixture.

Reviewer #3 (Remarks to the Author):

Overall, the study is presented in carefully planned and interpreted experiments and provides new and deeper insight into the integration of VOC signals in intra-plant and inter-plant immune signaling. What concerns me is that the authors somewhat over-interpret their results in terms of VOCs generating waves of immune signals in plant populations or within canopies. The plant-plant interaction experiments were conducted in closed desiccators, which allow accumulation of VOCs over a period of 24 h. While this experimental setup serves well for the purpose of studying VOCs as signals in SAR, it is uncertain whether a similar outcome would be observed in an open chamber or flow through system with lower VOC concentrations. The “boost” of volatile signals across multiple plants seems to me questionable in natural environments and the authors should revise their interpretation or show that the plant-plant interaction experiments are possible in an open flow system.

Response: We fully agree that it is important to demonstrate the potential physiological relevance of PTP signal transduction in open, ventilated, and dynamic systems (see also our answers to reviewer 1 and 2). In order to take this concern into account, we have carried out two additional experiments. (1) Vacuum desiccators with silicone tubes were connected to each other, allowing a regulated air flow from desiccator 1 with sender plants to desiccator 2 with receiver plants. The results depicted in the new main Figure 8 show that receiver plants reacted to PTP cues of infected sender plants with increased resistance to *Pst*. This response was dependent on the ability of the sender plants to emit monoterpenes. Also, propagation of PTP cues was observed in PTPTP experiments in which receiver 2 was exposed to sender 2 plants in the dynamic flow system (Figure 8). These data show that PTP propagation of innate immune cues between plants takes place in dynamic ambient air. (2) In addition, we conducted experiments in completely open systems with open glass vases and open (vacuum) desiccators (Figure S9). The results of both experiments, described on page 14, lines 1-22, strongly support our hypothesis that PTP propagation of immunity may be relevant in natural systems. Therefore, we have decided to stick to our working hypothesis that these volatile cues can be under self-reinforcing control and may also play an important ecological role in dense natural or agricultural plant

communities. We are confident that the data and conclusions in this revised version support our working hypothesis.

It appears from Figure 5F (ald1 response to VOCs from Col-0 WT) that other volatile compounds than monoterpenes are also involved in the immune response. I recommend that the authors test other possible WT-emitted VOC candidates for their role in SAR to address this question. I am also wondering to what extent VOC profiles vary between the different mutants. Could other VOCs than monoterpenes be affected in these mutants? Likewise, the ggpps12 mutant does not release monoterpenes; but, I would find a more thorough comparison of SAR-induced or plant sender-induced VOCs of this mutant with those of wild type plants helpful to exclude possible off-site effects or effects of the lack of a GPP pool on other terpene VOCs.

Response: We agree that the ald1 KO response to emissions from infected wildtype plants suggests the possible existence of additional volatile signals that may be relevant for PTP communication. With the help of further experiments and analyses, we have tried to address this concern. In the additional Figure S6, we now include data which show that ald1 mutant plants do not respond with enhanced resistance to the emissions of infected azi1-2 plants, although these emit volatile cues that trigger resistance in wildtype plants against Pst. Therefore, we used GC-MS analysis to compare the emission spectra of infected wild-type and azi1-2 mutant plants to possibly find additional compounds in the volatile PTP mixture that may act as volatile cues. However, the comparison showed no obvious differences. We also specifically checked the abundance of other volatiles that are known to play a role in plant-to-plant communication, such as lipoxygenase products (e.g. hexenol, hexenal, hexanyl acetate), isoprenoid sesquiterpenes, the benzenoid indole, and the aldehyde nonanal. However, we did not find any significant changes, suggesting that either the absence of pip in the ald1 mutant increased the sensitivity of the mutant to different monoterpene concentrations, or that additional volatile cues could not be detected by our VOC analysis. We discuss these results on page 9, line 31 to page 10, line 2 of the revised manuscript. In view of the results, we have adjusted in the text the possible role of pinenes as the sole cause of PTP proliferation of innate immunity. In the new version, we now point out that pinenes play an essential role in this process, "as part of a volatile mixture derived from plants" (page 2, line 14 and page 16, line 25).

Please note that the emission of volatile compounds in *A. thaliana* is very low and therefore all presented GC-MS data had to be integrated manually. It is also very important to accurately record

background emissions from the soil. We could not detect any terpenoids other than pinenes and camphene in the emissions of infected wildtype plants. Therefore, it is not possible to deduce possible side effects of reduced geranyl pyrophosphate (GPP) values in the *ggpps12* mutant on other volatile terpenes. We therefore agree with reviewer 3 that the results with the *ggpps12* mutant do not allow for a very clear statement. However, in order to further substantiate our hypothesis that monoterpenes are essential for PTP propagation of immunity, additional experiments with two independent T-DNA insertion alleles of terpene synthase 24 (*tps24*) were performed. Work by other groups has shown that recombinant TPS24 protein uses GPP as a substrate and synthesizes a monoterpene mixture containing pinenes and camphene in enzyme tests *in vitro*. In the new Figure S5 we now show that both *tps24* mutants exhibit reduced alpha-pinene emission after a SAR-inducing infection. Importantly, neither *tps24* mutant supported intra-plant systemic acquired resistance (Figure S7) or emitted defense-inducing PTP cues (as shown in the new Figure 5I). Hence, these additional data described on page 10, lines 10-27 strongly support a role of monoterpenes in intra- and inter-plant propagation of innate immunity.

REVIEWERS' COMMENTS:

Reviewer #1 (Remarks to the Author):

The authors have done a great job in revising the paper, and have addressed all my comments in a satisfactory manner.

One detail that should be clarified in the final version of the paper is the use of the Grubb test to identify and remove outliers. Removal of outliers needs to be justified carefully (an extreme value per se is not sufficient for removal, as it may be part of the biology of the system) and needs to be documented in detail (ideally by providing the raw data).

Signed: Matthias Erb

Reviewer #2 (Remarks to the Author):

The authors has adequately addressed my comments. I would like to endorse this manuscript for publication.

I recommend that the authors move Figure S9 to main text.

Reviewer #3 (Remarks to the Author):

The manuscript by Tan et al. has been under previous review. The authors included several experiments to respond to my concerns and those by the other reviewers, which have substantially improved the quality of the study. While not all questions could be answered based on experimental limitations, the authors show, in my opinion, convincingly that VOCs, even at low concentrations, mediate cascades of SAR-dependent plant immunity.

I was a bit surprised that the authors could only detect alpha-pinene and no other monoterpene in the tps24 T-DNA insertion lines even though TPS24 makes other major monoterpene products. This might indicate that TPS24 only contributes partially to alpha-pinene emissions in leaves or that the other TPS24 products are not made in vivo or are further metabolized. Do the authors have RT-qPCR data that show the transcript abundance of TPS24 in the T-DNA insertion lines? Is the gene still expressed at lower levels, which would support the partial reduction of alpha-pinene emissions compared to wild-type? It would be worthwhile adding those results to the supporting information.

Reviewer #1 (Remarks to the Author):

The authors have done a great job in revising the paper, and have addressed all my comments in a satisfactory manner.

One detail that should be clarified in the final version of the paper is the use of the Grubb test to identify and remove outliers. Removal of outliers needs to be justified carefully (an extreme value per se is not sufficient for removal, as it may be part of the biology of the system) and needs to be documented in detail (ideally by providing the raw data).

Signed: Matthias Erb

Response: Dear Matthias, thanks for your constructive comments (both prior and current). Grubb's outlier test detected outliers in 4 data sets (1 outlier in each), which were statistically significant. Their removal from the respective data sets in most cases assured normal distribution of the remaining data, allowing straight-forward ANOVA testing. In the revised manuscript, outlier removal is delineated in the appropriate figure legends (Fig. 1A, Fig. 4A, Suppl. Fig. 2B, and Suppl. Fig. 5). The full data, including outliers is included in the source data file associated with this manuscript; outliers that were removed to ensure normal data distribution (Fig. 1A, Suppl. Fig. 2B, and Suppl. Fig. 5) or improve figure readability (Fig. 4A) are highlighted in grey in the source data file.

Data presented in Fig. 4B and Supplementary Fig. 5 were defined as ND if the average alpha- or beta-pinene emissions were below background. In the remaining data sets, individual data points that were below background were also considered ND and thus not plotted in the bars. These data points were, however, included in the statistical analysis of the data. This is now also delineated in the appropriate figure legends.

With this, we are confident to have provided full data disclosure as per your request.

Reviewer #2 (Remarks to the Author):

The authors has adequately addressed my comments. I would like to endorse this manuscript for publication.

I recommend that the authors move Figure S9 to main text.

Response: Thank you for this and prior suggestions; the change you suggest here was made!

Reviewer #3 (Remarks to the Author):

The manuscript by Tan et al. has been under previous review. The authors included several experiments to respond to my concerns and those by the other reviewers, which have substantially improved the quality of the study. While not all questions could be answered based on experimental limitations, the authors show, in my opinion, convincingly that VOCs, even at low concentrations, mediate cascades of SAR-dependent plant immunity.

I was a bit surprised that the authors could only detect alpha-pinene and no other monoterpene in the *tps24* T-DNA insertion lines even though *TPS24* makes other major monoterpene products. This might indicate that *TPS24* only contributes partially to alpha-pinene emissions in leaves or that the other *TPS24* products are not made in vivo or are further metabolized. Do the authors have RT-qPCR data that show the transcript abundance of *TPS24* in the T-DNA insertion lines? Is the gene still expressed at lower levels, which would support the partial reduction of alpha-pinene emissions compared to wild-type? It would be worthwhile adding those results to the supporting information.

Response: Thank you for this and prior constructive comments. Please note that only alpha-pinene emissions are shown in *tps24-1* and *tps24-2*, because this was the only monoterpene we could detect in these particular experiments, also in the emissions of infected wt plants. Because recombinant *TPS24* protein converts GPP to a mixture of different monoterpenes, including alpha-pinene, beta-pinene, and camphene (Chen et al., 2003), we considered alpha-pinene in this case as representative for monoterpenes. This is included on page 9, line 4 of the revised text.

Because monoterpene emissions after infection were low in both *tps24* mutants, we did not pay much attention to checking *TPS24* transcript levels in these lines. This was now done and the data are included in the new Supplementary Figure 7. Unexpectedly, we detected *TPS24* transcripts at near-wt levels in both mutants. Therefore, we confirmed the exact insertion sites of the T-DNA insertions in both mutants. To do this, we generated PCR products on genomic DNA of both lines (2 plants each) across the T-DNA-gene border. Sequencing of the resulting PCR fragments revealed that insertion sites were within the fourth intron and 3' UTR of *TPS24*, respectively. Considering low monoterpene emissions after infection of both *tps24* mutant alleles, the data suggest misregulation of functional *TPS24*. These data are described on page 8, line 34 until page 9, line 3 of the revised text; the data are shown in the new Supplementary Figure 7 and the sequences revealing the T-DNA insertion sites are included in the source data file associated with this manuscript.

To account for the above data, we adjusted the conclusion about the possible role of *TPS24* in monoterpene biosynthesis on page 9, lines 5-7 to: '...concluded ... that *TPS24*, possibly together with other *TPS*, contributes to monoterpene emissions after infection.'